# A dual role of prestimulus spontaneous neural activity in visual object recognition

Ella Podvalny [1], Matthew W. Flounders [1], Leana E. King[1], Tom Holroyd[2] & Biyu J. He [1,3]

Vision relies on both specific knowledge of visual attributes, such as object categories, and general brain states, such as those reflecting arousal. We hypothesized that these phenomena independently influence recognition of forthcoming stimuli through distinct processes reflected in spontaneous neural activity. Here, we recorded magnetoencephalographic (MEG) activity in participants ($N = 24$) who viewed images of objects presented at recognition threshold. Using multivariate analysis applied to sensor-level activity patterns recorded before stimulus presentation, we identified two neural processes influencing subsequent subjective recognition: a general process, which disregards stimulus category and correlates with pupil size, and a specific process, which facilitates category-specific recognition. The two processes are doubly-dissociable: the general process correlates with changes in criterion but not in sensitivity, whereas the specific process correlates with changes in sensitivity but not in criterion. Our findings reveal distinct mechanisms of how spontaneous neural activity influences perception and provide a framework to integrate previous findings.

[1] Neuroscience Institute, New York University School of Medicine, New York, NY 10016, USA. [2] Magnetoencephalography Core Facility, National Institute of Mental Health, Bethesda, MD 20892, USA. [3] Departments of Neurology, Neuroscience & Physiology, and Radiology, New York University School of Medicine, New York, NY 10016, USA. Correspondence and requests for materials should be addressed to E.P. (email: ellapodvalny@gmail.com) or to B.J.H. (email: biyu.jade.he@gmail.com)

Perception does not rely on sensory input alone, but is also informed by previously acquired knowledge of sensory environment. For example, object recognition—that is, experiencing a percept of an object from physical properties of the stimulus projected to the retina—is informed by knowledge of object categories, which is supported by dedicated anatomical structures (for reviews see Grill-Spector and Weiner[1]; Logothetis and Sheinberg[2]). Previous sensory experience shapes these anatomical structures[3,4] and, consequently, modulates the neural activity these structures support[5,6]. Furthermore, dynamic content-specific information about previously experienced sensory environment was observed in neural activity even in the absence of sensory input[7–11]. An intriguing hypothesis suggests that such information provides continuous expectations about the content of forthcoming sensory stimuli[7,9,12,13]. However, it is currently unknown whether content-specific spontaneous neural activity plays a role in perception or, alternatively, is a mere byproduct of the underlying anatomical structure with no functional consequence.

In a separate vein, neuromodulatory systems, which regulate general arousal state, influence both the neural representation of sensory stimuli[14–16] and perceptual decisions about these stimuli[17,18]. Such regulation is associated with modulation of large-scale spontaneous brain activity[19–21]. Thus, spontaneous neural activity may also influence object recognition in a non-content-specific manner.

Research that has previously examined the influence of spontaneous activity on visual perception (e.g.[22–25]) has typically not tested whether such influence is content-specific or not. It has been shown that perception of ambiguous bistable images (such as Rubin's face-vase illusion) is biased by pre-stimulus activity in category-selective visual regions[26], but such biases were not tested in their ability to facilitate perceptual processing. Therefore, it remains unknown whether and how different processes existing in spontaneous neural activity—content-specific fluctuations constrained by local anatomical structure vs. non-content-specific fluctuations of global arousal—influence perception of forthcoming stimuli.

We hypothesized that object recognition can be influenced by two complementary spontaneous neural processes acting according to: (1) General model: pre-stimulus brain states influence recognition of a forthcoming stimulus regardless of the stimulus content; and (2) Specific model: pre-stimulus brain states facilitate recognition in a category-specific manner, such that each object category is associated with a pre-stimulus brain state that is optimal for recognizing that particular category (see illustration in Fig. 1a and Methods). We report evidence supporting the existence of two distinct spontaneous neural processes behaving according to the General and Specific models. Furthermore, we observe a double dissociation in the effects of these two processes on subjects' recognition behavior as characterized by signal detection theory, suggesting distinct mechanisms utilized by these two spontaneous processes.

## Results

**Paradigm and behavior**. In order to understand the influence of pre-stimulus neural activity on visual object recognition, we designed an experimental paradigm where object stimuli are presented at the threshold of subjective recognition. To this end, we conducted an adaptive thresholding procedure, whereby image contrast was titrated to reach a 50% subjective recognition rate for each subject (see Fig. 1d and Methods). In order to calculate the subjective recognition rate, the subjects were instructed to report whether they see an object ("yes"/"no"), such that even if the object appears unclear or noisy they should respond "yes",

but if they see nothing or only low-level features, such as lines or cloud-like abstract patterns, they should respond "no". The present paradigm is analogous to threshold-level visual detection tasks using simple low-level stimuli[25,27,28], but with important differences in stimulus type (Gabor patches vs. objects) and the definition of threshold (stimulus visibility vs. object recognition). We did not use a mask to render the stimulus invisible[29–31] because the mechanism of masking involves disruption of stimulus processing and thus may "mask" the influence of pre-stimulus activity[32].

Stimuli included four common visual object categories: faces, animals, houses and manmade objects (Fig. 1c). Participants' task was to report the category of an image presented and their recognition experience (yes/no, Fig. 1b). They were instructed to report the object category regardless of their recognition experience and, in cases of unrecognized images, to make a genuine guess (four-alternative choice discrimination). The stimulus set included real and scrambled images, where the scrambled images were created by phase-shuffling a randomly chosen real image from each category to preserve category-specific low-level image features (see Methods). Because scrambled images did not include an object stimulus, they were used as "catch trials" to determine the subjects' baseline tendency to give positive responses to a question about their recognition experience. Each object category included five unique real images and one scrambled image and each image was repeated 15 times across the experiment. The pre-stimulus interval varied randomly from trial to trial between 3 and 6 s in order to prevent stimulus timing predictability and the stimuli were presented in a randomized order to prevent category predictability.

Participants reported $44.9 \pm 3.5\%$ (mean ± s.e.m., $N = 24$) of real images as recognized (i.e., percentage of "yes" reports), which did not differ from the intended recognition rate of 50% (Fig. 1e, Wilcoxon signed-rank test, $p = 0.2$). The recognition rate of scrambled images was $17.2 \pm 3.3\%$, significantly below the recognition rate of real images (Fig. 1e, Wilcoxon signed-rank test, $p = 1.8 \times 10^{-5}$) and significantly above zero (Wilcoxon signed-rank test, $p = 4.0 \times 10^{-5}$). The subjects were instructed to reply positively to a question about their recognition experience only if they could detect an object in the stimulus presented; therefore, a "yes" report of a scrambled image constitutes a "False Alarm" (Fig. 1d). There are two possible interpretations of the significant False Alarm rate: (1) the subject perceived an object even though no object was present, and (2) the subject did not actually perceive an object but pressed a wrong report button, either by accident or because they forgot what they saw. The report of recognition experience is inherently subjective; therefore, we cannot discern between these two possibilities.

We also examined the effect of trial history on subjective recognition rate. First, we found that a "yes" report on a previous trial (as compared to a "no" report) correlates with an increase of $7.2 \pm 1.6\%$ in subjective recognition rate (Supplementary Fig. 1b, Wilcoxon signed-rank test, $p = 8.3 \times 10^{-4}$). Similar, but weaker, effect of $4.1 \pm 1.3\%$ increase in recognition rate was found for two trials back (Wilcoxon signed-rank test, $p = 6.6 \times 10^{-3}$) but not further (Wilcoxon signed-rank test, $p > 0.05$ for 3, 4, and 5 trials back). Second, we examined the effect of the objective stimulus category in the previous trial on subjective recognition rate. We found that having two subsequent trials of the same (as compared to different) objective stimulus category does not influence subjective recognition rate (Supplementary Fig. 1c, Wilcoxon signed-rank test, $p = 0.98$).

We used signal detection theory (SDT)[33] approach to characterize the recognition decision criterion (i.e., tendency to report "yes" regardless of whether a real or scrambled image is presented) and recognition sensitivity (i.e., estimated distance

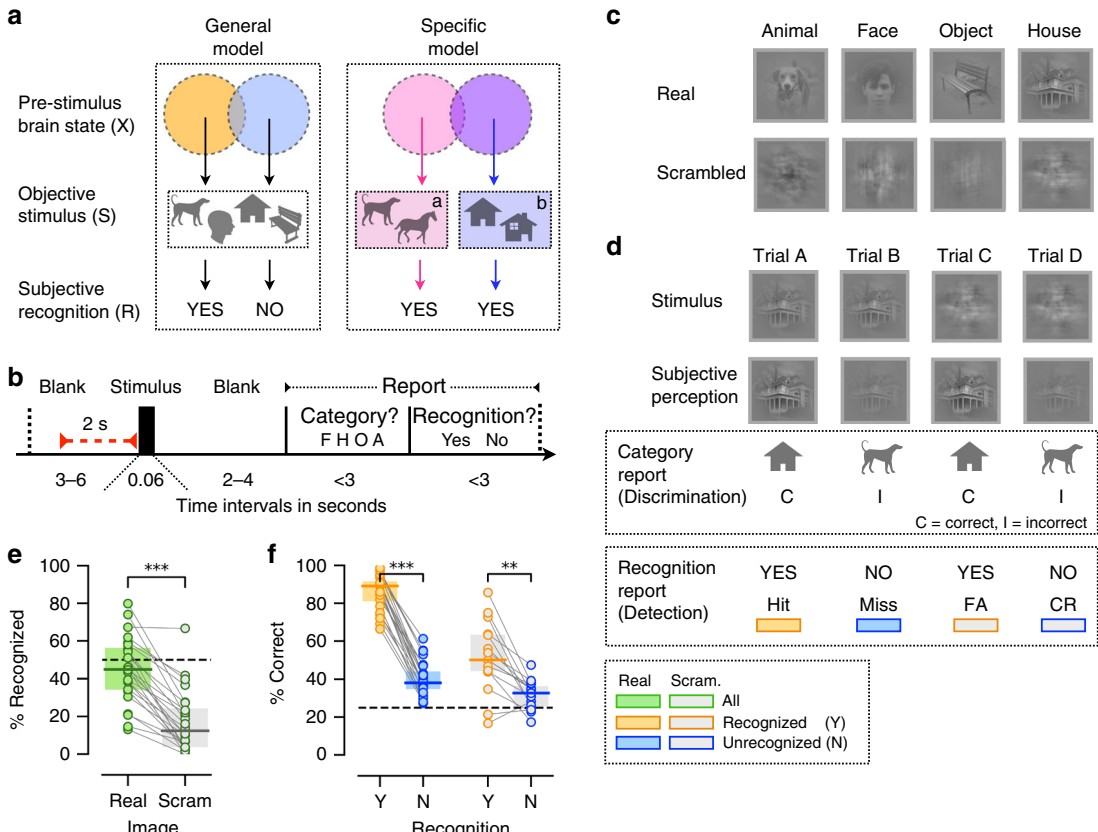

**Fig. 1** Hypothesis, experimental paradigm and behavioral results. **a** Hypothesis illustration. General model: The stimulus (S) has a preferred pre-stimulus brain state (orange) that increases the probability of recognition experience (R = yes), regardless of objective stimulus category. Specific model: Pre-stimulus brain states facilitate recognition in a category-specific manner: the optimal brain state for recognizing a stimulus from category $a$ differs from the optimal state for recognizing a stimulus from category $b$. **b** Paradigm timeline. Each trial began with a fixation cross presented on a blank screen. Next, a brief stimulus was presented at a contrast determined by the participant's individual recognition threshold. Subjects were asked to report the stimulus category and their recognition experience. Neural activity recorded in a 2-s window prior to stimulus onset was used in neural data analyses (red bar). F/H/O/A correspond to Face/House/Object/Animal. **c** Example real and scrambled images from each stimulus category. **d** Illustration of trial types. For each subject, identical real and scrambled stimuli (top) may be reported as recognized on one trial and not recognized on another. When real stimulus is reported as recognized, it is considered a "Hit" because it contains an object (bottom). The accuracy of category discrimination in scrambled-image trials is determined based on the original stimulus category used to generate the scrambled image **e** Percentage of trials reported as recognized for real and scrambled images. Dashed line indicates intended threshold-level recognition rate for real images (Wilcoxon signed-rank test). **f** Accuracy of category report in recognized and unrecognized trials of real (N = 24) and scrambled images (Wilcoxon signed-rank test, N = 15, only subjects reporting more than five scrambled-image trials as recognized are included). In all figures, boxplot center line depicts the median and edges indicate the quartiles; *p < 0.05, **p ≤ 0.01, ***p ≤ 0.001, n.s.: not significant. Source data are provided as a Source Data file

between distributions of sensory representations for real and scrambled images). The subjects' recognition reports had significant sensitivity ($d' = 1.02 \pm 0.11$, Wilcoxon signed-rank test, $p = 1.8 \times 10^{-5}$) and a positive (i.e., conservative) decision criterion ($c = 0.66 \pm 0.11$, Wilcoxon signed-rank test, $p = 10^{-4}$), in line with previous studies[27,28,34,35]. We will later describe how these two components of object recognition behavior are influenced by pre-stimulus brain states.

We next characterized categorization behavior in recognized and unrecognized trials. The categorization accuracy of real images was $86.5 \pm 1.8\%$ in recognized trials and $40.1 \pm 1.8\%$ in unrecognized trials, which were significantly different ($p = 1.8 \times 10^{-5}$, $N = 24$, Wilcoxon signed-rank test, Fig. 1f and Supplementary Fig. 1a). Consistent with previous studies[28,36,37], categorization accuracy in unrecognized trials was still significantly above the chance level of 25% ($p = 1.8 \times 10^{-5}$, $N = 24$, Wilcoxon signed-rank test). We observed a similar trend in trials in which scrambled images were shown: first, categorization accuracy was $51.9 \pm 4.7\%$ in recognized and $31.6 \pm 1.9\%$ in

unrecognized trials, significantly above chance in both groups (Wilcoxon signed-rank test, $p = 1.2 \times 10^{-3}$ and $p = 7.6 \times 10^{-3}$, respectively), suggesting that low-level image features that are distinct between categories[38] contribute to categorization behavior; second, the categorization accuracy was significantly higher in recognized than unrecognized trials (only subjects with more than five scrambled images in each group were included, $N = 15$, Wilcoxon signed-rank test, $p = 1.5 \times 10^{-3}$), suggesting that low-level image features potentially contribute to False Alarms.

Below we describe the results of testing the General and Specific models of the influence of pre-stimulus activity on object recognition (Fig. 1a). During task performance, subjects' brain activity was recorded by a 275-channel whole-head MEG system. We defined pre-stimulus brain state as the activity pattern obtained by averaging the MEG signal in a two-second window before stimulus presentation in each trial (red bar, Fig. 1b). This approach did not require frequency-domain filtering of the data and therefore avoids potential filter-induced signal leakage from post- to pre-stimulus time window. Furthermore, slow

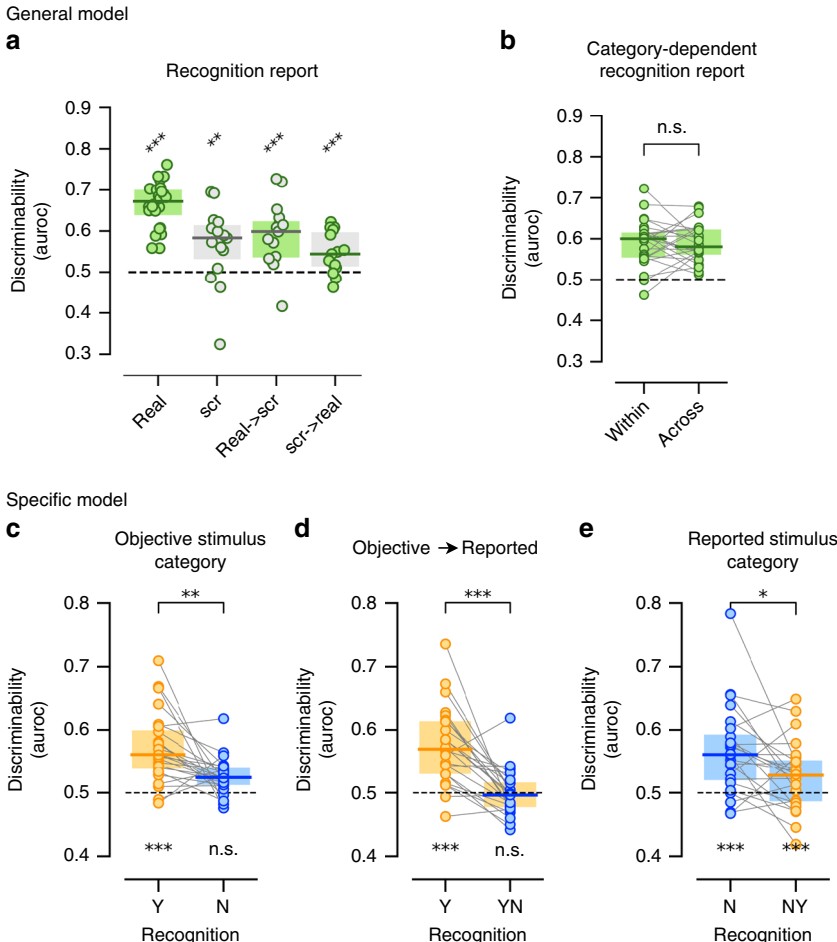

**Fig. 2** Pre-stimulus activity contains predictive information about recognition of a forthcoming stimulus. **a** Discriminability of recognition report (Y/N) calculated using a logistic regression model fit to pre-stimulus activity patterns, for real and scrambled images separately (1st and 2nd bar, $N = 24$ and $N = 15$, respectively). The 3rd and 4th bars show the results of decoding across image type, where the model was fit using real images and tested using scrambled images and vice versa ($N = 15$). **b** Discriminability of recognition report (Y/N) calculated using a model trained and tested within each objective stimulus category and with the model performance averaged across categories (1st bar) and a model trained on trials from one category and tested on trials from another category and with the performance averaged across category pairs (2nd bar). **c** Discriminability of the forthcoming objective stimulus category calculated using a logistic regression model fit to pre-stimulus activity patterns, for recognized and unrecognized trials separately. **d** Discriminability of a logistic regression model fit to pre-stimulus activity patterns that was trained to predict objective stimulus category in recognized trials and tested in its ability to predict reported stimulus category in recognized trials (1st bar, using a leave-one-out cross-validation scheme) and unrecognized trials (2nd bar). **e** Discriminability of the reported stimulus category calculated using a logistic regression model fit to pre-stimulus activity patterns. 1st bar: trained and tested on unrecognized trials. 2nd bar: trained on unrecognized trials and tested on recognized trials. In **b**–**e**, only trials in which real images are presented are used ($N = 24$). In all panels, dashed line indicates the theoretical chance level. Whenever training and testing used the same trial group (i.e., not cross-condition decoding), a leave-one-out cross-validation scheme was used. Label permutation was used to assess significance of each model (i.e., each bar) and Wilcoxon signed-rank test was used to asses difference between models. Source data are provided as a Source Data file

fluctuations in ongoing brain activity recorded by electrical field potentials have been shown to reflect coherent fluctuations in brain networks[39] and influence detection of forthcoming stimuli[40,41]. These activity patterns were used to fit multivariate logistic regression models (see Methods) that predict recognition experience ("yes"/"no") and objective or reported stimulus category (face/object/animal/house). The models' performance was assessed by using a leave-one-out cross-validation scheme and estimating the area under the receiver-operator curve (AUROC, see Methods and Supplementary Code 1).

**Testing the General model**. To test whether pre-stimulus activity influences recognition according to the General model (Fig. 1a), we fit a logistic regression model based on pre-stimulus activity

patterns and used it to generate single-trial predictions of recognition reports ("yes"/"no"). For trials wherein real images were presented, we found that pre-stimulus brain states indeed predicted the reported recognition of upcoming stimuli (AUROC = $0.66 \pm 0.01$, $p < 0.001$, label permutation test, $N = 24$; Fig. 2a, first bar; model activation pattern analysis is shown in Supplementary Fig. 2a, b).

Since subjective recognition also occurs in a substantial fraction of scrambled-image trials (Fig. 1e), we performed a similar analysis using brain states preceding scrambled images. For this analysis, only subjects who reported recognition in more than five scrambled-image trials were included ($N = 15$, see methods). Model performance was significantly above chance for scrambled-image trials as well (AUROC = $0.57 \pm 0.02$, $p = 6 \times 10^{-3}$, label permutation test; Fig. 2a, second bar). We next tested

whether the models that were trained on real and scrambled images utilized similar features of pre-stimulus brain states to predict the recognition reports. To this end, we examined whether the model trained on real-image trials had the power to predict recognition reports in scrambled-image trials, and vice versa. We found significant AUROC scores of $0.59 \pm 0.02$ and $0.54 \pm 0.01$ in both cross-decoding analyses ($p < 10^{-3}$, label permutation test; Fig. 2a, 3rd and 4th bars). Thus, a shared spontaneous neural process influences subjective recognition of both real and scrambled images.

We next tested whether the pre-stimulus activity patterns influencing recognition identified here are specific to each objective stimulus category or shared across categories. The latter scenario would fit with the General model. To distinguish between these two possibilities, we compared the performance of cross-validated models that were fit and tested using trials from the same objective stimulus category (within-category) with models that were fit using trials from one category and tested using trials from a different category (across-category). There was no difference in performance between within-category and cross-category models (AUROC = $0.59 \pm 0.012$ and AUROC = $0.59 \pm 0.010$, respectively, Wilcoxon signed-rank test, $p = 0.841$; Fig. 2b), suggesting cross-category generalization of the influence of pre-stimulus activity patterns on recognition. Together, these results provide strong evidence for the General model by showing the existence of a spontaneous neural process that influences whether an upcoming stimulus is reported as recognized, regardless of the stimulus content.

**Testing the Specific model.** Our hypothesis for the Specific model (Fig. 1a) states that pre-stimulus brain states can influence object recognition in a category-specific manner. In this case, the brain state that facilitates recognition of a stimulus from category $a$ differs from the brain state that facilitates recognition of a stimulus from category $b$. This hypothesis predicts that in recognized trials, pre-stimulus activity patterns contain information about the objective category of forthcoming stimuli. Even though stimuli from different categories were presented in a randomized order (i.e., stimulus category was objectively unpredictable throughout the experiment) such prediction is possible because the model predicts conditional probability (see Methods). Intuitively, the Specific model suggests that if the brain makes a valid prediction about the category of a forthcoming stimulus (i.e., predicted category equals objective stimulus category), then this stimulus is more likely to be recognized; thus, in recognized trials we should be able to identify the objective stimulus category from the pre-stimulus brain state better than chance.

In order to test this hypothesis, we fit a logistic regression model to predict objective stimulus category at the single-trial level based on pre-stimulus brain states and tested the model using a leave-one-out cross-validation scheme. We found that model discriminability for objective stimulus category was significantly above chance in recognized trials (AUROC = $0.57 \pm 0.012$, label permutation test, $p < 10^{-3}$, Fig. 2c, model activation pattern analysis is shown in Supplementary Fig. 2c, d), but not unrecognized trials (AUROC = $0.52 \pm 0.006$, label permutation test, $p = 0.16$). Furthermore, objective stimulus category discriminability based on pre-stimulus brain states was higher in recognized compared to unrecognized trials (Wilcoxon signed-rank test, $p = 2 \times 10^{-3}$). Therefore, in recognized trials alone, the category of a forthcoming stimulus was predicted correctly more often than chance by a model fit to pre-stimulus activity patterns, suggesting that spontaneous activity influences object recognition also in a content-specific manner as predicted by the Specific model.

According to the Specific model, predictions of objective stimulus category are made by spontaneous neural activity before stimulus onset; when such a prediction is valid, the stimulus is more likely to be recognized (see Illustration in Fig. 3a). We next asked whether this same spontaneous neural process also influenced subjects' categorization behavior in real images. If this is the case, a logistic regression model trained to decode objective stimulus category in recognized trials (i.e., identifying neural activity behaving according to the Specific model, Fig. 2c, 1st bar) should also have predictive power in decoding subjects' reported category in both recognized and unrecognized trials. In recognized trials, objective stimulus category is equal to reported stimulus category in >85% of trials; thus, a model trained to predict objective stimulus category would be able to predict reported stimulus category trivially. We first confirmed that this is indeed the case (AUROC = $0.57 \pm 0.013$, label permutation test, $p < 10^{-3}$; Fig. 2d, 1st bar). By contrast, in unrecognized trials, the same model had no predictive power regarding reported stimulus category (AUROC = $0.5 \pm 0.01$, label permutation test, $p = 0.44$; Fig. 2d, 2nd bar). Therefore, the spontaneous neural process that facilitates subjective recognition in a category-specific manner does not directly influence subjects' forced discrimination of object category in real images.

To gain a better understanding of the category report system, we tested whether there is any information in pre-stimulus brain states that influenced subjects' reported category. Given the high redundancy between objective stimulus category and reported stimulus category in recognized trials, we focused on unrecognized trials here. We found that a logistic regression model trained on pre-stimulus activity was indeed able to predict the reported stimulus category better than chance in unrecognized trials (AUROC = $0.57 \pm 0.01$, label permutation test, $p = 10^{-3}$, Fig. 2e, 1st bar). Interestingly, this model also showed significant discriminability for reported stimulus category when trained on unrecognized trials but tested on recognized trials (AUROC = $0.53 \pm 0.01$, label permutation test, $p = 10^{-3}$; Fig. 2e, 2nd bar). Together, these results suggest that the spontaneous neural process that facilitates object recognition in accordance with the Specific model is distinct from the process biasing the category report.

**Behavioral consequences of General and Specific processes.** So far, we have shown that spontaneous neural activity can influence object recognition in manners consistent with the General and the Specific model. Next, to shed more light on the nature of this effect, we examined how fluctuations in these neural processes on a single-trial level influence two orthogonal aspects of recognition behavior as defined according to SDT: criterion (c), which captures the tendency of a subject to respond "Yes" (i.e., to report subjective recognition of object) regardless of whether the image contains an object, and sensitivity (d′), which captures the ability of a subject to distinguish between real and scrambled images.

To assess the behavioral consequences of pre-stimulus neural processes acting according to the General and Specific models, we extracted decision variables produced by logistic regression decoders in the form of predicted probabilities (see Fig. 3a and Supplementary Fig. 3). For the General process, we used the decision variable produced by a model fit to activity patterns preceding real images (Fig. 2a, 1st bar) that corresponds to predicted probability of recognition, P(recognition). For the Specific process, we used the decision variable produced by a model fit to activity patterns preceding recognized real images (Fig. 2c, 1st bar) that corresponds to predicted probability of an objective stimulus category, P(category). Using the probabilities predicted on each trial, we split all trials into two groups for each

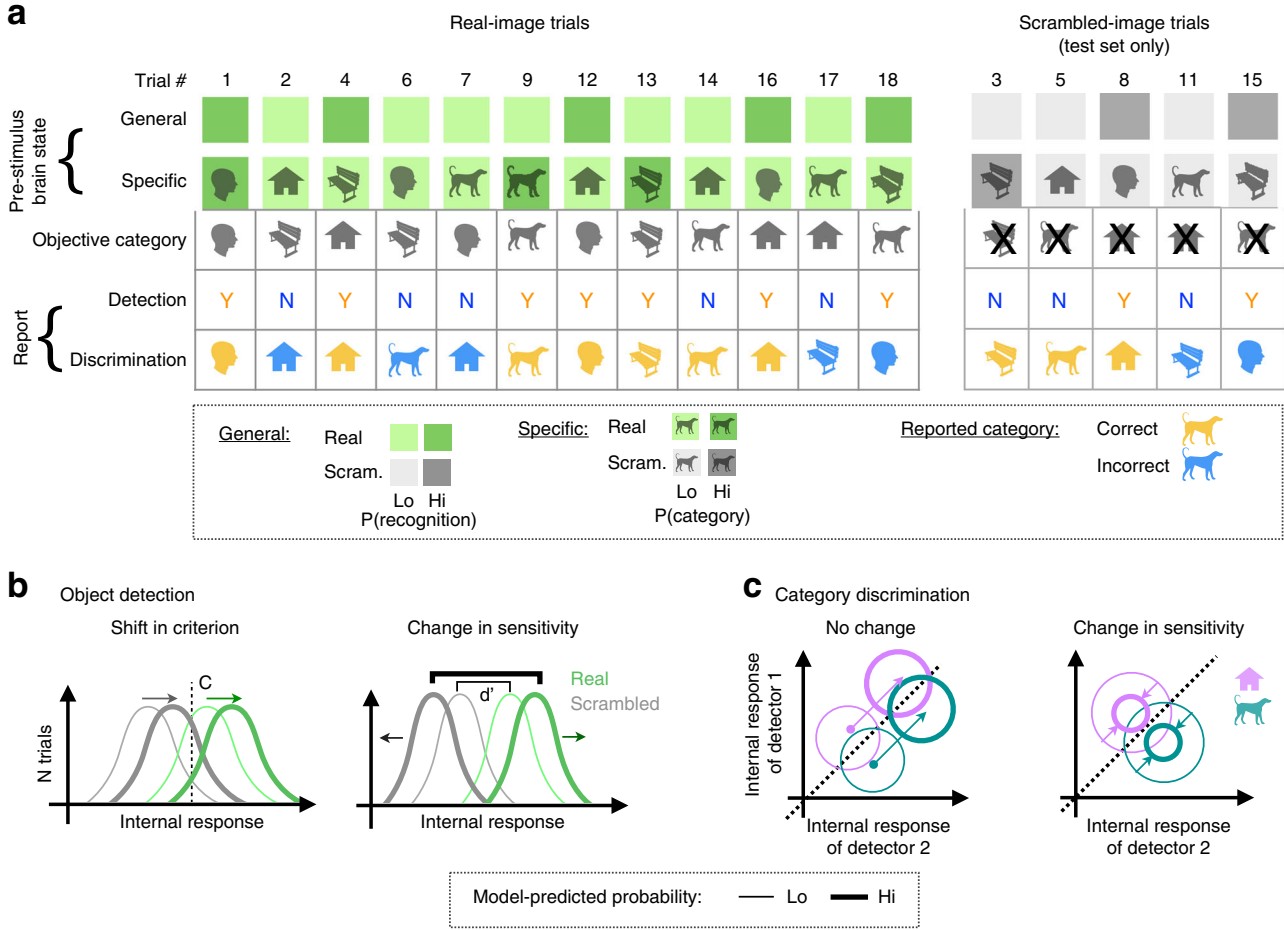

**Fig. 3** Illustration of behavioral consequences of the General and Specific spontaneous neural processes. **a** Illustration of single-trial predictions made by the General and Specific neural process. According to the General model, pre-stimulus brain states influence the probability of recognizing a forthcoming stimulus, P(recognition), regardless of its category. According to the Specific model, pre-stimulus brain states contain predictive information about objective stimulus category, P(category), which translates into a higher probability of recognition if valid (i.e., category predicted with the highest probability matches the objective stimulus category). The two shades of green/gray depict high (hi) and low (lo) probability predicted by each model trained in real-image trials. **b** The two models fit within real-image trials may correlate with shifts in criterion and changes in sensitivity of object detection. The thick and thin lines depict high and low probability predicted by each model. **c** The two models fit within real-image trials may correlate with changes in category discrimination of both real and scrambled-image trials. The thick and thin lines depict high and low probability predicted by each model

process (Fig. 3, "hi" and "lo" groups). For the General process, we defined a high-probability group as trials where the predicted probability of the image to be recognized was higher than the probability of that image to be unrecognized; the remaining trials were placed in the low-probability group. For the Specific process, we defined a high-probability group as trials where the predicted probability of the objective forthcoming stimulus category was higher than that of any other category (i.e., the objective forthcoming stimulus category was predicted correctly by the model using information present in pre-stimulus activity); all other trials were placed in the low-probability group. Note, all models for this analysis were trained using real-image trials only, while the probabilities were predicted for scrambled-image trials and real-image trials through cross-validation.

The results so far indicate that both ongoing brain processes, General and Specific, influence subjective recognition, but do not inform on whether such effects stem from a shift in criterion, a change in sensitivity, or both. In SDT framework, recognition experience report ("yes"/"no") can be interpreted as a signal detection task, where "signal" refers to an object stimulus—present in real images and absent in scrambled images. Similarly,

the categorization task is a four-alternative force choice task, where discrimination sensitivity can be operationally measured by percent correct of category reports[33]. Figure 3b, c illustrates potential influences ongoing brain states may have on subjects' recognition (Fig. 3b) and categorization (Fig. 3c) reports.

To test these hypotheses, we first calculated the recognition rates in groups of trials defined by the above approach. For the General model, the recognition rate in real-image trials (i.e., "Hit Rate") was $0.56 \pm 0.03$ in high and $0.36 \pm 0.03$ in low probability groups. The recognition rate was higher when the predicted probability of recognition by the General process is high compared to low (Fig. 4b, Wilcoxon signed-rank test, $p = 1.82 \times 10^{-5}$, $N = 24$). For the Specific model, the Hit Rate was $0.50 \pm 0.04$ in high and $0.43 \pm 0.04$ in low probability groups with a significant difference between groups (Fig. 4b, Wilcoxon signed-rank test, $p = 6.7 \times 10^{-4}$, $N = 24$). These results stem from our earlier observations that there are spontaneous neural processes influencing subjective recognition according to both the General and the Specific model. In trials where scrambled images were presented, the models that were trained using real-image trials only predicted recognition rates (i.e., "False Alarm Rate") of $0.23 \pm 0.03$ and

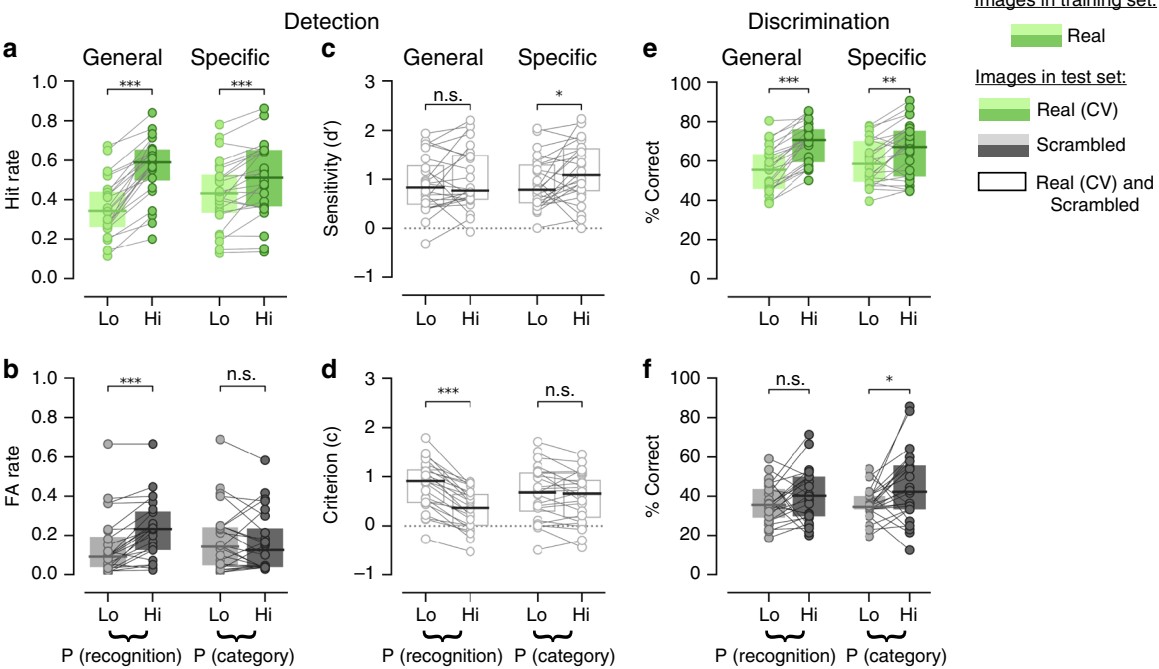

**Fig. 4** The predictions made by the General and Specific models differentially bias the SDT metrics. **a–d** Show the metrics of detection (based on subjective recognition reports: "yes" or "no") and **e–f** show the metrics of discrimination (based on subjective stimulus category reports: "Face", "House", "Object", or "Animal"). **a** Recognition rates as a function of predictions made by the General and Specific (cross-validated, CV) neural process in real-image trials (Hit Rate). **b** Same as **a** but for scrambled-image trials (False Alarm rate) while the models were fit in real-image trials. **c** SDT sensitivity (d′) as a function of predictions made by the General and Specific neural process. **d** SDT criterion (c) as a function of predictions made by the General and Specific neural process. **e** Percent of correct subjective stimulus category reports for the two models in real-image trials (cross-validated, CV). **f** Percent of correct subjective stimulus category reports for the two models in scrambled-image trials, while the model was fit in real-image trials only. Wilcoxon signed-rank test was used to assess the difference between the groups in **c, b, e, f** and paired t-test to assess the difference between groups in **c, d** ($N = 24$). See Supplementary Figure 3 for details on how trials were split into high and low probability groups. Source data are provided as a Source Data file

$0.15 \pm 0.03$ for the high and low probability groups according to the General model, and $0.17 \pm 0.03$ and $0.18 \pm 0.03$ according to the Specific model (Fig. 4b). The recognition rate varied according to the General process (Wilcoxon signed-rank test, $p = 1.83 \times 10^{-4}$, $N = 24$) but not the Specific process (Wilcoxon signed-rank test, $p = 0.43$, $N = 24$).

Since recognition rates are influenced by both criterion and sensitivity, the interpretation of the above results alone is ambiguous. To resolve this, we next calculated SDT metrics within each group of trials. According to the General model, criterion (c) was $0.34 \pm 0.08$ and $0.83 \pm 0.1$ and sensitivity (d′) was $1.01 \pm 0.13$ and $0.9 \pm 0.12$ in high and low probability groups. We found that criterion (c), but not sensitivity (d′), was influenced by the pre-stimulus activity patterns associated with the General model (Fig. 4c, d, paired t-test, $p = 1.63 \times 10^{-9}$ and $p = 0.28$, respectively, $N = 24$). Thus, the neural process underlying the General model influences recognition by shifting the decision criterion in the liberal direction. According to the Specific model, criterion was $0.57 \pm 0.1$ and $0.66 \pm 0.11$ and sensitivity was $1.16 \pm 0.13$ and $0.92 \pm 0.11$ in high and low probability groups. Unlike for the General model, sensitivity (d′), but not criterion (c), was influenced by the pre-stimulus activity patterns associated with the Specific model (Fig. 4c, d, paired t-test, $p = 4.66 \times 10^{-2}$ and $p = 0.12$, respectively, $N = 24$). Thus, the neural process underlying the Specific model influences recognition by enhancing recognition sensitivity.

We further tested whether the two spontaneous brain processes influence the accuracy of subjects' categorization reports. The percentage of real images categorized correctly was $68.77 \pm 2.12\%$ and $55.31 \pm 2.33\%$ according to the General model and $65.67 \pm$

$2.92\%$ and $59.14 \pm 2.32$ according to the Specific model in high and low probability trial groups, with both models having a significant influence (Fig. 4e, Wilcoxon signed-rank test, General: $p = 1.82 \times 10^{-5}$; Specific: $p = 3.91 \times 10^{-3}$, $N = 24$). Given that scrambled images also contain category-specific low-level features (Fig. 1f), it is conceivable that ongoing brain processes may also influence subjects' categorization of scrambled images. To test this hypothesis, we used the models trained on real-image trials and tested their effects on subjects' categorization accuracy in scrambled-image trials. The percentage of scrambled images categorized correctly was $40.39 \pm 2.8\%$ and $36.49 \pm 2.1\%$ according to the General model and $44.71 \pm 3.77\%$ and $36.1 \pm 1.64$ according to the Specific model in high and low probability trial groups. Notably, only the Specific process significantly biased categorization accuracy in scrambled image trials (Fig. 4f, $p = 3.45 \times 10^{-2}$ for Specific, $p = 0.14$ for General, $N = 24$). These results indicate that both spontaneous processes influence category discrimination decisions made about forthcoming real images, but only the Specific process also enhances category discrimination performance on scrambled images.

In sum, we observe a double dissociation in how spontaneous neural processes related to the General and the Specific model shape object recognition (captured by "Yes" vs. "No" answers): the former shifts decision criterion whereas the latter enhances sensitivity of object detection.

**Trial-history in General and Specific processes**. We examined the relationship between predictions made by the two models and the subjective recognition reports made on the preceding or

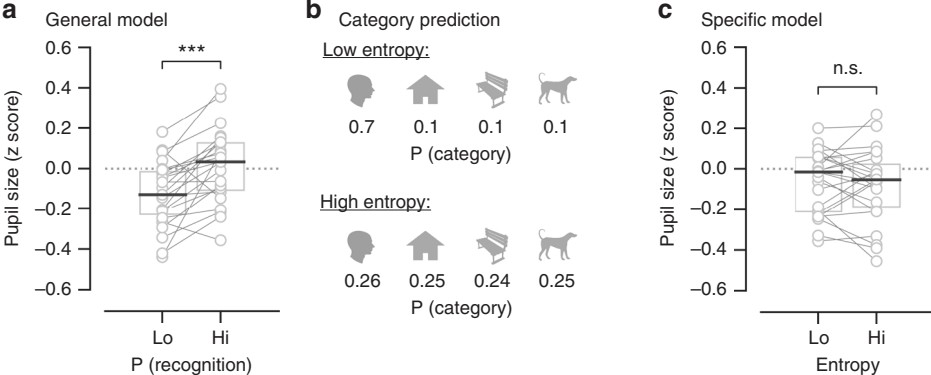

**Fig. 5** General, but not specific, spontaneous neural process is related to pupil-linked arousal. **a** Pre-stimulus pupil size as a function of recognition probability predicted by the General neural process. **b** Illustration of stimulus category prediction with low and high entropy (top and bottom panels, respectively), reflecting certainty of prediction on two different trials. **c** Pre-stimulus pupil size as a function of the uncertainty (quantified by entropy) in stimulus category prediction made by the Specific neural process. Wilcoxon signed-rank test was used to assess the difference between the groups ($N = 23$, one subject excluded due to eye-tracking failure). Source data are provided as a Source Data file

ensuing trial. For the General model, trials split by probability of subjective recognition predicted on the previous trial were not different in recognition rate (Wilcoxon signed-rank test, $p = 0.63$) nor did the subjective recognition report on a previous trial influence the predicted probability by the model (Wilcoxon signed-rank test, $p = 0.63$). For the Specific model, similarly, the model-predicted category matched the objective stimulus category presented on the previous trial at a rate of $25.7 \pm 0.5\%$, not higher than the chance level of 25% (Wilcoxon signed-rank test, $p = 0.13$). In addition, the model-predicted category did not influence the recognition rate of the next trial whether it matched the objective stimulus category in this next trial or not (Wilcoxon signed-rank test, $p = 0.33$). Furthermore, there was no significant across-trial temporal autocorrelation in the decision variables predicted by the two models. Thus, while our behavioral result showed a trial history effect in subjective recognition behavior (Supplementary Fig. 1B), its link to the spontaneous processes uncovered herein is unclear at present.

**The General process is related to pupil-linked arousal**. Lastly, we tested whether the spontaneous neural processes acting according to the General and Specific models correlate with trial-to-trial arousal fluctuations as indicated by pupil size. For the General neural process, we used the same procedure to split trials into high and low probability groups as in the above analysis (Figs. 3 and 4). We compared the mean pupil size across a two-second pre-stimulus window between the two groups, and found that higher predicted probability of recognition correlated with a larger pupil size ($-0.13 \pm 0.03$ and $0.02 \pm 0.04$ for low and high-probability groups, Wilcoxon signed-rank test, $p = 5.9 \times 10^{-4}$, $N = 23$, Fig. 5a). Thus, the General neural process is related to fluctuations in arousal.

For the Specific neural process, we hypothesized that higher arousal may lead to more specific (i.e., certain) predictions of stimulus category. In each trial, the Specific spontaneous neural process generates a predicted probability for each stimulus category (face, house, object, animal), which sum to 1 (see illustration in Fig. 5b). The prediction is completely certain if the predicted probability for one stimulus category is 1 and the others 0. On the other hand, assigning a probability of 0.25 to each category would mean that the model is entirely uncertain. We utilized the measure of entropy to quantify uncertainty of category prediction made by Specific spontaneous neural process

in each trial, which ranged from 0 to 1, corresponding to the aforementioned two extreme cases. Trials were split into two groups using a median-split based on entropy. Pupil size did not vary with the uncertainty of category prediction (Fig. 5c, $-0.08 \pm 0.03$ and $-0.07 \pm 0.03$ for high and low entropy groups, Wilcoxon signed-rank test, $p = 0.63$, $N = 23$). Thus, the General, but not Specific, pre-stimulus neural process correlates with pupil-linked arousal.

## Discussion

In sum, we uncovered two separate spontaneous neural processes that influence visual object recognition. First, we observed a general process, which influences recognition regardless of the stimulus content. Second, we identified a specific process, which makes spontaneous predictions about the category of forthcoming stimuli and influences recognition in a content-specific manner. Only the general process correlated with pupil size, which reflects arousal fluctuations. The two processes result in doubly dissociable effects on object recognition behavior: the general process correlates with a shift in decision criterion with no effect on sensitivity, while the specific process correlates with a change in sensitivity with no effect on criterion. These findings support a dual-model framework for the role of spontaneous neural activity in sensory perception, which offers new perspectives for interpreting existing results and points to new directions for future research.

It is well established that pre-stimulus neural activity can influence the detection of simple sensory stimuli[23,24,42,43]. However, previous studies have not tested whether this influence is content-specific or not. Several considerations suggest that previously observed effects of pre-stimulus activity on sensory detection were largely non-content-specific. First, previous studies have reported that pre-stimulus activity, including α oscillation power and neuronal firing rates, influences the criterion but not the sensitivity of sensory detection[35,44]. Further, pre-stimulus fMRI activity in visual cortex influences false alarm rates of stimulus detection[45], which is akin to recognition of scrambled images in our experiment and is related to a shift in criterion. Second, observations of spontaneous activity in large-scale attentional networks influencing sensory detection[42,43] are likely related to fluctuations in arousal and sustained attention. Given our findings showing that the general, but not the specific, spontaneous neural process influences recognition criterion and

correlates with arousal, these previous findings are likely related to the non-content-specific general process identified herein.

Spontaneous (i.e., non-light-mediated) pupil size fluctuations reflect locus coeruleus neuronal activity and norepinephrine modulations of neural activity in the neocortex[46,47]. Recent results suggest that spontaneous pupil fluctuations correlate with widespread cortical activity in the default-mode network[19], frontoparietal attention networks[48] and sensory cortices[19,48–50]. It has also been shown that pupil size correlates with attention[48,51] and, together, arousal and attention appear to exert overlapping effects on cortical activity and sensory encoding[50,52]. We observed that the spontaneous neural process acting in accordance with the General model correlates with trial-to-trial fluctuations in pre-stimulus pupil size; therefore, it is likely supported by global brain states and large-scale neural activity.

Theories have postulated that spontaneous, content-specific neural activity may facilitate perceptual processing[9,12,13,53]. Here, we provide the first experimental evidence supporting this hypothesis. Specifically, we report that a valid category prediction —that is, when the category predicted by pre-stimulus activity matches the objective stimulus category—enhances human subjects' recognition sensitivity of object stimuli without influencing recognition criterion. While it has been reported that previously experienced sensory environment shapes content-specific spontaneous activity patterns[8,54,55], we now show that such activity patterns directly influence perception of forthcoming stimuli in humans. Thus, our finding establishes that content-specific spontaneous neural activity indeed plays a role in perception, by carrying predictive information about future stimuli content and enhancing perceptual sensitivity when such information is valid.

Our findings may also shed light on potential neural mechanisms involved in Bayesian theories of perceptual inference. Bayes formula can be used to describe object perception as an integration of image feature reliability, embodied by likelihood, and the prior probability distribution. For instance, Kersten et al.[56] suggested that "under ambiguous conditions, the visual system is forced to guess, but it can make intelligent guesses by biasing its guesses toward typical objects or interpretations"[56]. Fiser et al. suggested that prior knowledge, such as knowledge about typical objects, could be represented as "samples" in spontaneous neural activity[7,57]. It was unknown, however, how a process of "biasing" toward typical objects is implemented on a single-trial level and in what way it influences perception. The spontaneous category-specific activity patterns we observed could represent individual "samples" from prior knowledge. We found that these patterns facilitate the perception of forthcoming stimuli by enhancing recognition sensitivity, suggesting that valid predictions increase the strength of content-specific signals, which results in a larger distance between neural representations of object and scrambled noise stimuli.

It is possible that the spontaneous neural processes reported herein partially reflect spontaneous fluctuations in cognitive processes such as attention and expectation. While in the present study we did not explicitly manipulate these cognitive processes, spontaneous fluctuations in neural activity are suggested to correlate with spontaneous fluctuations in cognition[58]. Furthermore, top-down cognitive processes influence object recognition[59,60]. It is also possible that bottom-up visual signals simply encounter the ongoing neural state of the local networks along the visual processing hierarchy, which in turn affects processing of these signals. These possibilities of the relationship between spontaneous neural processes and top-down/bottom-up stimulus processing deserve future investigation.

The current study does not provide substantial insights into the anatomical underpinnings of the general and specific spontaneous neural processes. Source reconstruction of pre-stimulus MEG data in the context of tasks is challenging because pre-stimulus activity is typically used to estimate the noise covariance matrix for guiding source reconstruction of post-stimulus activity. Our finding of a double dissociation between the general and the specific processes points to non-overlapping underlying mechanisms. Future studies employing intracranial recordings or fMRI should help with providing anatomical insight into the phenomena uncovered herein by testing whether the general and specific processes are supported by distinct anatomical areas.

Specifically, we suggest to test in future studies whether the general ongoing process is supported by brain areas involved in the control of arousal and the specific ongoing process by areas encoding object representations. First, recent reports suggest that it is possible to track arousal fluctuations using resting-state fMRI signals in widespread cortical and subcortical regions[19–21,61]. Yet, a direct influence of these arousal-related modulations on sensory processing remains to be shown. Second, it remains to be tested whether the specific ongoing process is supported by brain areas specialized in object processing, such as the ventral temporal cortex (VTC). Recent studies demonstrated the contribution of VTC to object category decoding from whole-brain MEG signals by employing novel computational approaches to combine MEG and fMRI data[62,63]. Thus, reactivation of stimulus-triggered patterns in spontaneous VTC activity may underlie the specific ongoing process uncovered herein—a hypothesis that warrants future investigation. In sum, our findings provide new hypotheses to be tested in future studies.

Much is known about how the brain represents visual objects. In natural settings, however, object recognition does not rely on sensory input alone but is regulated by ongoing brain dynamics to a large extent. For example, a percept of an object can emerge in the complete absence of sensory stimuli (as in dreams and hallucinations) and, on the other hand, salient stimuli can go unnoticed (e.g., during inattentional blindness). Our findings reveal a heretofore unknown dual role of spontaneous neural activity and shed light on how the brain implements object recognition beyond mere extraction of sensory features. These findings offer a new framework for future research where the two ongoing neural processes (content-specific and non-content-specific) we identified may have dissociable impacts on other perceptual and cognitive functions as well.

## Methods

**Participants.** All participants ($N = 25$, 15 females, mean age 26, range 22–34) provided written informed consent. The experiment was approved by the Institutional Review Board of the National Institute of Neurological Disorders and Stroke (protocol #14-N-0002). The participants were right-handed, neurologically healthy, and had normal or corrected-to-normal vision. One enrolled participant decided to stop the experiment after finishing one experiment block due to discomfort and is not included in data analyses. We chose a sample size similar to those used in recent published MEG studies on perception and cognition conducted with healthy human volunteers.

**Experimental stimuli.** Images were selected from four categories—faces, animals, houses, and objects (Fig. 1c). The images, selected from public domain labeled photographs or from Psychological Image Collection at Stirling (PICS, http://pics.psych.stir.ac.uk/), were resized to $300 \times 300$ pixels and converted to grayscale. The actual experimental stimuli from PICS are not available for commercial use; therefore, the images shown in Figures were downloaded from https://www.pexels.com/ and are presented to demonstrate the outcome of image processing procedure described below. The pixel intensities, ranging from 0 (black) to 255 (white), were normalized by removing the mean and dividing by standard deviation and filtered using a 2-D Gaussian smoothing kernel with a standard deviation of 1.5 pixels and $7 \times 7$ pixels size (imgaussfilt, MATLAB). Five unique images were included in each category, resulting in 20 unique real images in total. Scrambled images were created by shuffling the 2-D Fourier transformed phase of one randomly chosen image from each category. The edges of the images were gradually brought to background intensity by multiplying the image intensity with a Gaussian window with a standard deviation of 0.2. Stimuli were presented using the Psychophysics

Toolbox[64] in MATLAB via a projector with a 60-Hz refresh rate onto a screen. Stimulus size was 8° in diameter. The stimuli were presented gradually from 0.01 to threshold intensity during ~66.7 ms (four video frames).

**Experimental trial structure.** Each trial started with a gray background (Fig. 1b). After a blank screen of a random duration between 3 and 6 s (following an exponential distribution), a stimulus was presented. Then another blank screen with a duration randomly chosen between 2 and 4 s (following an exponential distribution) was presented. The luminance of the blank screens was equal to the background luminance of the stimulus screen. The first blank period ensured that the subject could not predict the onset of the stimulus. Each trial ended with two sequential questions (see section Task and instructions below). Subjects indicated their answers to the questions via a right-hand keypad. A central fixation cross was present at all times except during response prompts and subjects were instructed to fixate whenever the fixation cross is present.

**Task and instructions.** First, subjects were instructed to report image category. When the image was not recognized, subjects were instructed to guess randomly (four-alternative forced choice) with an emphasis on making a genuine guess to the best of their ability. The response mapping, indicated by the order of words "Face"/ "House"/"Object"/"Animal" on the screen, was randomized across trials. Second, subjects were instructed to report their recognition experience with a "yes" or "no" choice, where "yes" was defined as "something that makes sense in the real world". Subjects were instructed to report "yes" even if the image was not entirely clear or they saw only part of the image, such as only the eyes and ears of an animal without knowing exactly what animal it was. Subjects were instructed to report "no" if they did not see anything at all or if they saw random noise patterns. Subjects were not informed of the use of scrambled images.

**Threshold image contrast definition.** We used an adaptive staircase procedure "QUEST"[65] to find image contrast ($c$) yielding a recognition rate of 50% (proportion of "yes" responses to the second question). The image pixel intensity, $I$, at a given contrast, $c$, was calculated as:

$$I(c) = b(I_{scaled} * c + 1) \quad (1)$$

where $b$ is the background intensity (set to a constant value of 127) and scaled pixel intensities ($I_{scaled}$) were obtained by rescaling the image pixel intensities between −1 and 1. As a result, the lightest pixel value in the image was equal to $I_{max} = b(1 + c)$ and the darkest $I_{min} = b(1 - c)$. Therefore, we defined the contrast of a presented image as:

$$c = \frac{I_{max} - I_{min}}{2b} \quad (2)$$

which ranged between 0 and 1.

Subjects performed the QUEST procedure under conditions identical to the subsequent main task, except that the trial timing was faster (750 ms pre-stimulus interval and 1 s post-stimulus interval). The initial contrast of each image in the QUEST procedure was determined by the mean of threshold contrasts obtained in three pilot subjects who were not part of the present study. For each subject in the present study, the QUEST procedure included 120 trials, split into three individual staircase procedures; the median among the three final threshold contrasts was selected for the main task.

**Experiment structure.** First, each subject performed an adaptive staircase procedure to determine the threshold image contrast. After a short break, the main experiment began, where the trials were shown repeatedly with an identical stimulus contrast at the subject's individually determined threshold while MEG signals were continuously recorded. The main experiment had 360 trials with 300 real-image trials and 60 scrambled-image trials. Each unique image was repeated 15 times. The images were presented in a random order; therefore, the image category was unpredictable by the subject. The trials were split into 10 experimental blocks containing 36 trials each. Each block ended with a self-paced break period.

**Eye-tracking and pupil size preprocessing.** During the experiment, subjects' eye position and pupil size were continuously monitored and recorded using an Eye-Link 1000 + system, in the binocular mode with a sampling rate of 1000 Hz. Blinks were detected by identifying the time points where pupil diameter of the right eye dropped by a threshold of 3.6 measurement units; blink onset was defined as 40 ms before crossing the threshold and blink offset was defined as 40 ms after (due to tendency for MEG artifacts to occur within this time window). For further analyses, pupil diameter measurements were averaged in a time window of 2 s before stimulus onset, while excluding the time points affected by blinks. No frequency-domain filtering was applied. One subject was excluded from pupil size analysis due to inferior quality of data on all experiment blocks.

**MEG data acquisition and preprocessing.** MEG data were recorded at a sampling rate of 1200 Hz using a 275-channel scanner (CTF, VSM MedTech). Before and after each block, the head position of the subject was measured using coils placed on the ear canals and the bridge of the nose. Between blocks, the head position of the subject was measured with respect to the MEG sensor array using coils placed on the left and right preauricular points and the nasion, and the subject self-corrected their head position to the same position recorded at the start of the first block using a custom visual-feedback program [written by TH, inspired by (Stolk et al., 2013)[66] in order to minimize head displacement across the experiment. All MEG data samples were corrected with respect to the refresh delay of the projector (measured with a light sensor). MEG data were preprocessed using Python and the MNE toolbox[67] (version 0.17.1). Three dysfunctional sensors were removed from all analyses. Independent component analysis (ICA) was performed on each block to remove eye-movement, blink, cardiac and movement-related artifacts. The linear trend was removed from each experimental block. No frequency-domain filtering was applied in order to avoid artifactual signal bleeding from the post-stimulus signal into the pre-stimulus period. Finally, for each trial and each sensor the recorded MEG data were averaged in a 2-sec time window before stimulus onset. Based on eye-tracking data, trials in which a blink occurred during stimulus presentation were excluded, resulting in 12.0 ± 2.3 (mean ± s.e.m.) rejected trials for 20 participants. Loss of eye-tracking occurred in four subjects in several experiment blocks, hence no trials were rejected in those subjects based on blinks.

**Formal hypotheses.** Let $\mathbb{X}$ be a set of brain states preceding the appearance of stimuli from category in set $\mathbb{S}$ and let $\mathbb{R}$ be the set of subjective recognition reports

$$\mathbb{X} = \{\mathbf{x}_1, \mathbf{x}_2, \dots \mathbf{x}_n\} \quad (3)$$

$$\mathbb{S} = \{face, object, animal, house\} \quad (4)$$

$$\mathbb{R} = \{yes, no\} \quad (5)$$

A pre-stimulus brain state $\mathbf{x} \in \mathbb{X}$ is a vector each dimension of which signifies an attribute of a brain state (for example, activity recorded with one MEG sensor). A brain state may influence recognition report $r \in \mathbb{R}$ of stimulus $s \in \mathbb{S}$ in two manners:

General model hypothesis: A subset of brain states $\mathbb{X}'$ exists ($\mathbb{X}' \subset \mathbb{X}$), such that the probability (Pr) of recognizing stimulus $s \in \mathbb{S}$ is higher than not recognizing it if the brain is in state $\mathbf{x}' \in \mathbb{X}'$ at the time of stimulus arrival:

$$\Pr(r = yes | \mathbf{x}', s) > \Pr(r = no | \mathbf{x}', s) \quad (6)$$

where the probabilities of recognition outcomes on a given trial sum to 1: $\Pr(r = yes) + \Pr(r = no) = 1$

This hypothesis can be tested using a logistic regression model:

$$\log\left(\frac{\Pr(r = yes | \mathbf{x}, s)}{\Pr(r = no | \mathbf{x}, s)}\right) = \boldsymbol{\beta}\mathbf{x} \quad (7)$$

$$\frac{\Pr(r = yes | \mathbf{x}, s)}{1 - \Pr(r = yes | \mathbf{x}, s)} = e^{\boldsymbol{\beta}\mathbf{x}} \quad (8)$$

$$\Pr(r = yes | \mathbf{x}, s) = \frac{1}{1 + e^{-\boldsymbol{\beta}\mathbf{x}}} \quad (9)$$

A model that is able to discriminate between the two groups of brain states $\mathbb{X}'$ and $\sim\mathbb{X}'$, with a decision boundary at $\boldsymbol{\beta}\mathbf{x} = 0$, would present a supporting evidence for this hypothesis. Note, it is required that the stimulus will be sometimes recognized and sometimes not in order to fit and test the model. In a more general case, however, certain stimuli can have no plausible pre-stimulus brain state (for example, extremely weak stimulus that can never be recognized). On the other extreme, some stimuli can be unrecognized only rarely and have many preferred brain states (for example, a salient face stimulus can be unrecognized in extreme states of inattention).

Specific model hypothesis: A subset of brain states $\mathbb{X}_a$ exists ($\mathbb{X}_a \subset \mathbb{X}$) such that the probability of recognizing stimulus $s_a \in \mathbb{S}$ is higher than the probability of recognizing stimulus $s_b \in \mathbb{S}$ arriving when the brain is in state $\mathbf{x}_a \in \mathbb{X}_a$:

$$\Pr(r = yes | \mathbf{x}_a, s_a) > \Pr(r = yes | \mathbf{x}_a, s_b) \quad (10)$$

By definition of conditional probability:

$$\Pr(r | \mathbf{x}, s) = \frac{\Pr(r, \mathbf{x}, s)}{\Pr(\mathbf{x}, s)} = \frac{\Pr(s | \mathbf{x}, r)P(\mathbf{x}, r)}{\Pr(\mathbf{x}, s)} \quad (11)$$

and assuming the stimulus category is independent of pre-stimulus brain state:

$$\Pr(\mathbf{x}, s) = \Pr(\mathbf{x})\Pr(s) \quad (12)$$

the hypothesized inequality (10) can be written as:

$$\frac{\Pr(s_a | \mathbf{x}_a, r = yes)\Pr(\mathbf{x}_a, r = yes)}{\Pr(\mathbf{x}_a)\Pr(s_a)} > \frac{\Pr(s_b | \mathbf{x}_a, r = yes)\Pr(\mathbf{x}_a, r = yes)}{\Pr(\mathbf{x}_a)\Pr(s_b)} \quad (13)$$

Reducing equal terms and given that the different categories were presented with equal probabilities:

$$\Pr(s_a|\mathbf{x}_a, r = \text{yes}) > \Pr(s_b|\mathbf{x}_a, r = \text{yes}) \quad (14)$$

The existence of the inequality above can be tested with a logistic regression model formulated as:

$$log\left(\frac{\Pr(s_a|\mathbf{x}, r = \text{yes})}{\Pr(s_b|\mathbf{x}, r = \text{yes})}\right) = \boldsymbol{\beta_a}\mathbf{x} \quad (15)$$

$$\Pr(s_a|\mathbf{x}, r = yes) = \frac{1}{1 + e^{-\boldsymbol{\beta_a}\mathbf{x}}} \quad (16)$$

Thus, if a preferred brain state for stimulus $s_a$ processing exists, we should be able to predict the objective stimulus category using logistic regression model fit to pre-stimulus brain activity in recognized trials.

**Multivariate pattern analysis**. Multivariate pattern analyses were performed with scikit-learn package for Python[68] (version 0.20.2, see Supplementary Code 1 for details). We used penalized logistic regression model with L2 norm regularization (C = 1) and a leave-one-out cross-validation scheme. We used Coordinate Descent (CD) algorithm to fit the binary models and Newton's Method algorithm to fit the multi-class models. To quantify the predictive power of binary models (i.e., predicting the recognition report: "yes"/ "no"), area under the receiver-operator curve (AUROC) was estimated. ROC was constructed by using the class probabilities predicted by a model and shifting the discrimination threshold. An AUROC of 0.5 indicates no predictive power, and a value of 1 indicates perfect predictive power. To quantify the performance of multi-class models (i.e., predicting object category: face/house/object/animal), we calculated receiver-operator curve for each class versus others and calculated the area under the averaged curve.

The statistical significance of AUROC scores was estimated using group-level label permutation tests. For each model tested, we repeated the analysis 1000 times with labels shuffled across trials in the training set and re-computed the AUROC value for each permutation—this procedure was used to estimate the data-driven chance-level distribution of AUROC values[69]. The actual average AUROC across subjects was compared to the average AUROC values of 1000 permutations and the p-value was computed as the fraction of permuted scores that exceeded the actual score.

Probability of forthcoming recognition experience and probability of forthcoming stimulus/report category were calculated for each trial using the logistic regression model:

$$\Pr(Y = y|\mathbf{x}) = \frac{1}{1 + e^{-\boldsymbol{\beta_y}\mathbf{x}}} \quad (17)$$

where $\mathbf{x}$ is the pre-stimulus brain state, $y$ is the predicted class label ("yes"/"no" for recognition and face/object/animal/house for category), and $\boldsymbol{\beta_y}$ is the fitted model parameters. Pre-stimulus brain state, $\mathbf{x}$, was calculated using MEG data recorded from $M$ sensors and averaged across time in a 2-s window before stimulus onset (see MEG data preprocessing above for more details) and, for every experimental trial, $\mathbf{x}$ constituted an $M$-dimensional vector. In this scenario, each MEG sensor constitutes a model feature, and $\boldsymbol{\beta_y}$ is an $M$-dimensional vector of weights. All sensors were used to fit the models except the analyses in Supplementary Fig. 2b, d where separate groups of sensors were used to fit and test each model. Non-zero model weights ($\boldsymbol{\beta_y}$) together maximize class-specific information and suppress noise or distracting signals. In order to examine to which extent each sensor drives the model performance we calculated "activation patterns" as: $\mathbf{A} = \boldsymbol{\Sigma_x}\boldsymbol{\beta_y}$, where $\boldsymbol{\Sigma_x}$ signifies data covariance[70] for each model performing better than chance. The group-level activation patterns were calculated using the median of subject-level activation patterns (presented in supplementary Fig. 2).

In analyses that involved calculating AUROC for decoding of the recognition report in scrambled images (Fig. 2a, 2nd–4th bars), only subjects that reported at least 5 scrambled images as recognized were included ($N = 15$). This minimal number of trials was required in order to fit the model (i.e., "training trial set") and to derive a meaningful interpretation of decoder discriminability from a separate set of trials (i.e., "testing trial set"). All 24 subjects were included in decoding analyses that did not rely on scrambled-image trials (Figs. 2b–e, 4).

**Signal detection theory analysis**. We calculated Signal detection theory (SDT) metrics for detection and discrimination.

Detection: we calculated measures of sensitivity ($d'$) and bias ($c$) following standard SDT analysis[33] using subjective reports of recognition (i.e., responses to the second question: "yes" or"no"). $d'$ indicates the ability to discriminate between real images containing objects and scrambled images that do not contain objects but preserve low-level features of the object images. It is computed by subtracting the Z-transformed False Alarms Rate (FAR) from the Z-transformed Hit Rate (HR):

$$d' = Z(\text{HR}) - Z(\text{FAR}) \quad (18)$$

where Z is an inverse normal cumulative distribution function.

$c$ criterion represents the tendency to make "yes" reports to indicate recognition, regardless of whether the stimulus is a real or a scrambled image and is

computed as follows:

$$c = -\frac{1}{2}(Z(\text{HR}) + Z(\text{FAR})) \quad (19)$$

We implemented Macmillan & Kaplan correction[71] of FAR = 0 and HR = 1: the False Alarm rate was defined as the recognition rate in scrambled-image trials and was corrected to $\frac{1}{2N_{\text{scr}}}$ in the case of no FA trials, where $N_{\text{scr}}$ is the total number of scrambled-image trials; the Hit Rate was defined as the recognition rate of real-image trials and was corrected to $1 - \frac{1}{2N_{\text{real}}}$ in the case of Hit Rate equal to 1, where $N_{\text{real}}$ is the total number of real-image trials.

Discrimination: We calculated percent of correct responses as an operational measure of 4AFC (four Alternative Forced Choice) sensitivity using subjective category reports (i.e., responses to the first question: "Face", "House", "Object" or "Animal")[33,72].

**Entropy**. To quantify uncertainty in the predictions made by a logistic regression model, we used a measure of information entropy:

$$H(c) = -\sum_{i=1}^{n} \Pr(c_i) \log_n \Pr(c_i) \quad (20)$$

where $c$ is the predicted category, and $n$ is the number of categories we used in our experiment ($n = 4$).

**Statistics**. Potential group-level differences between paired groups of bounded variables (AUROC, %) were tested using nonparametric two-tailed Wilcoxon signed-rank tests as provided by the Scipy package[73]. One-sample Student t-test was used to assess differences in criterion and sensitivity. Repeated-measures ANOVAs were used to compare models across ROIs, we used the MNE implementation[67].

## Data availability
The datasets generated and analyzed during the current study are available from the corresponding authors on reasonable request. Source data files for all figures are provided.

## Code availability
We used publicly available open source software toolboxes written in Python to analyze our data. A demo script of a general workflow will be available with publication. Additional scripts or details about the code are available from the corresponding authors.

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

## Acknowledgements

This research was supported by National Science Foundation CAREER Award (BCS-1753218, to BJH) and NIH/NINDS Intramural Research Program Transition Funds

(to BJH). We thank Tashi Dolma for assistance in experimental paradigm development and testing, Enora Rice for helping with MEG data pre-processing, Max Levinson, Richard Hardstone and Thomas J. Baumgarten for comments on the paper. Graphic elements for figures were downloaded from https://publicdomainvectors.org/.

## Author contributions

B.J.H.: supervision, project administration, funding acquisition; B.J.H., and E.P.: conceptualization, methodology, writing—original draft; E.P.: software, formal analysis, visualization; M.F. and E.P.: experimental data collection, data curation; M.F., E.P., T.H.: investigation; L.E.K.: pupil data investigation and analysis; M.F., L.E.K., T.H.: Writing—review and editing; T.H.: resources.

## Additional information

**Competing interests:** The authors declare no competing interests.

