## [Peer Review File · Nature Communications]

Reviewers' comments:

Reviewer #1 (Remarks to the Author):

This manuscript reports the results of one MEG experiment (N=24) investigating the influence of pre-stimulus activity on perceptual decision making in the visual domain. Low-contrast images from four categories (faces, animals, houses, objects, and their scrambled counterparts) were presented for 66.67ms, and participants were instructed to a) report the image category, and b) rate their subjective "recognition experience" (yes or no, see below). The authors used the recorded MEG and behavioral data to test two models: a general model (pre-stimulus brain states "influence recognition of a forthcoming stimulus regardless of the stimulus content", l.48), and a specific model (pre-stimulus brain states "facilitate recognition in a category-specific manner", l.50). Using multivariate pattern analysis (MEG) in combination with SDT analysis (behavior), the authors find evidence in support of both models, and report that the two processes were doubly dissociable: "the general process shifted the criterion of recognition but had no effect on sensitivity, whereas the specific process enhanced sensitivity with no effect on criterion" (l.10). The authors conclude that their findings reveal an "unknown dual role of spontaneous neural activity in object recognition and provide a new framework for future research" (l.331). While the findings are potentially relevant and interesting, a couple of major points need to be addressed for a better understanding of these complex data.

Major points

1) I tend to disagree with one of the authors' concluding statements: "thus, sensory inputs merely bias intrinsic brain dynamics rather than acting as agents essential for perception" (l.330). The observation that even scrambled images (which only contain low-level features of the four image categories) shown for 66.67ms could still be successfully categorized above chance level (51.9%, l. 94) seems to be in conflict with that statement. If I am not mistaken, this observation rather shows that even weak sensory inputs clearly dominated the reported percepts.

2) With respect to the above-chance categorization performance (51.9%) for scrambled images, the authors write: "...suggesting that low-level image features ... contribute to categorization behavior and potentially to 'visual hallucination' (i.e., seeing a meaningful object out of meaningless patterns)" (l.97). I do not think that the above-chance categorization performance suggests that participants were "hallucinating" when confronted with the scrambled images. Rather, participants simply followed the instructions, as they were instructed to report the image category, and to guess it if necessary (l.66). So they used every low-level visual feature available to perform the categorization task. It remains unclear, however, how "vividly" they perceived the reported image category (i.e., the quality of their perception remains unknown). Therefore, I do not think that their performance can be compared to visual hallucinations (see also l.129). Furthermore, one potential weakness of the current paradigm seems to be that participants were not informed about the scrambled condition at all (l.515). It is not mentioned why this choice was made, and what its implications for participants' behavior might have been.

3) Related to my previous point, the authors do not address any potential weaknesses of their design, or alternative models that could explain their data. Overall, there seems to be too little room for doubt and nuance in this manuscript.

4) It is not mentioned how the sample size (N=24; for some analysis N=15, 22) was determined.

5) Terminology. The authors write: "Object recognition can happen in the complete absence of sensory stimuli (as in dreams and hallucinations)" (l.329). I would argue that "recognition" and "perception" cannot be used interchangeably, as it is done here. Similarly, the authors use the terms "recognition", "subjective recognition", "recognition rate", and "recognized trials" with respect to their second task, in which participants "were instructed to report their recognition experience" (l.511-516). The task is more like a subjective visibility rating (e.g., perceptual

awareness scale), and it should be consistently framed and labelled as such (e.g., "high" and "low" visibility, or "yes" and "no" trials, as opposed to "recognized" and "unrecognized" trials). The current terminology is somewhat misleading.

6) More details about the multivariate MEG analysis would be helpful and enable readers to fully understand the reported results. What MEG data exactly enters this analysis? How does this multivariate analysis relate to previous EEG/MEG studies on pre-stimulus activity and its influence on perception?

7) The results are analysed within a SDT framework, following "standard SDT analysis" (l.622). Low-contrast images from four categories (faces, animals, houses, objects; "real" images) and their scrambled counterparts were presented. These "real" and "scrambled" images were presented in random order, and they make up the total stimulus set. Thus, the participants' behavior in the two tasks (image categorization, and subjective rating of "recognition experience") is linked tightly to this stimulus set in its entirety. SDT analyses on subsets of these stimuli (e.g., "analyses that involved scrambled images", l.633) are therefore problematic at least, if not invalid (see lines 633-635).

8) As this is a categorization task with four categories, confusion matrices (for real and scrambled trials) would be informative, and could easily be added to the supplement. For example, it might turn out that faces were often correctly categorized, while the other categories were harder to categorize for participants.

9) I did not understand how the trials were split into two groups for each process (l.205). Was this done separately for each participant? Did this procedure result in the same number of trials in each group?

10) The results are primarily described in terms of "significant" ($p < .05$) and "not significant" ($p > .05$). It would be informative to learn more about the size of the effects, at least some descriptive stats could be added to the main text (e.g., when writing "d' was influenced by pre-stimulus activity patterns", or "d' was not influenced by...", the respective mean d' values could be added, in addition to the figures). Similarly, the important double dissociation between shifts of criterion and enhancement of perceptual sensitivity (lines 221-232) relies on "significant" and "non-significant" p-values alone, while it seems more appropriate to directly test the interaction (i.e., "model x hi/low" interaction, separately for d' and c, Figure 3).

Reviewer #2 (Remarks to the Author):

Review of: A dual role of spontaneous neural activity in visual object recognition

The present study investigated brain states that facilitate visual perception. The authors demonstrated, using magnetoencephalography (MEG) recordings, that certain brain-states enable better recognition of visual stimuli, presumably by lowering criterion threshold (i.e. higher propensity of a 'yes' answer), and furthermore that specific brain-states facilitate visual recognition of stimuli related to specific categories (man-made objects, animals, houses and faces).

The notion that the brain has intrinsic states that facilitate (or hamper) visual recognition is interesting and worthy of investigation. The paper is well-written and the study seems rigorous and well-executed.

Several comments that should be addressed are listed below for consideration:

1. Does the brain state that facilitate recognition of one category similar to the one facilitating perception of a stimuli from another category? Given that no source localization method seems to have been undertaken here, a precise answer for this question might not be possible, but the topoplots presented in supplemental Fig 2 suggest that a similar pattern of activation preceded all

of the four categories. How can the authors explain a similar activation for diverse categories in the specific model proposed?

2. Anticipations can make one category subsequently more or less recognizable, but expectations are not a brain "state." These are more different types of prepared-nesses that might stem from previous trials as well as the perceptual history with which the participants arrived to the experiment.

3. Spontaneous activation might result from persistence of some sort (especially because presentations did not use backward masking) and from top-down influences. A thoughtful discussion of top-down vs bottom-up influences on pre-stimulus "spontaneous" activity would be important and relevant.

4. Supplemental Fig 2 is a topoplot portraying the topography of activation prior to presentation of real-image trials. However, it is stated that this brain-state could also predict scrambled trials (as shown in Fig 2A). Was the pattern of activation in this general model similar in both instances?

5. The claim about doubly dissociable influences on object recognition (i.e. criterion and sensitivity) sounds a bit circular. The general and specific models were initially constructed by predicting correct recognition (a 'yes' answer for the general model or selection of a correct category in the specific model). In that case, isn't the association between the general model and criterion, and between the specific model and sensitivity, a by-product of this initial analysis? Some technical issues:

6. Given that eye-tracking was administered in this study, it is advisable to check for times where participants did not look at the screen and thus did not recognize the stimuli (i.e. instances in which the brain state could not predict visual perception simply because participants weren't looking).

7. What is the reason for a non-parametric testing across the analyses (instead of parametric tests)?

8. A scale indicating activation values is missing in supplemental Fig 2.

9. Under the methods section it is stated that one participant dropped out (thus N=23), yet in the results N=24 is indicated.

Reviewer #3 (Remarks to the Author):

This work represents an important advance in the understanding of the functional role of spontaneous brain activity. Similar to several other works, it studies the impact of pre-stimulus activity on subsequent perception of threshold stimuli using a prediction approach, however with the important novelty of separating fluctuations in a more general "content-unspecific" neural process and a category-specific neural process. As the current interest in spontaneous activity is very high (apparent e.g. in the countless resting state studies), this MEG study will be of broad interest. A few specific questions / suggestions follow.

The machine learning predictions are based on MEG sensor-level signals averaged over a 2s pre-stimulus period. In the discussion section the authors acknowledge the lack of topographical information due to methodological limitations. I am puzzled about what information is being harnessed by the model for the category-specific predictions? My assumption would be that brain activity dissociating upcoming perception across visual object categories would be localized in the ventral visual stream, an area that is hard to detect in MEG signals even if source-localization was possible.

Following a related line of thought, I am wondering whether the topographies for brain states preceding each of the four object categories provided in Supplementary Fig. 2C have similarity to the post-stimulus states evoked by each of the respective categories? The authors discuss that the spontaneous (pre-stimulus) category-specific activity patterns could represent samples or knowledge about typical objects. I interpret this to mean that the patterns that facilitate

perception of a certain category can be viewed as some degree of reactivation of neural patterns representing that category. Thus, they should have similarity to the post-stimulus evoked state of that category. In addition to a direct quantitative comparison of pre- and post-stimulus topographies, it would be interesting to see if training a model on the post-stimulus activity of the four categories could predict reported category when tested on pre-stimulus activity.

Regarding the Signal Detection Theory (SDT) application, I understand that both criterion c and sensitivity d' were estimated using hit and false alarm rates with respect to the second question, i.e. the question whether the subject recognized the object. This is in contrast to the approach of the rest of the paper up to this point, in which the general (category-insensitive) process was targeted using the 2nd question, while the category-specific process was targeted using the first question, i.e. which of four categories the subject saw. If I understand correctly, this SDT approach thus assumes that reporting "Yes" to the recognition question (2nd question) on a scrambled image is considered a false alarm, but the behavioral results showed that subjects were actually successful in using low-level features of the scrambled images to correctly "recognize" the category far above chance. This implies that answering "Yes" on a scrambled trial is not actually a false alarm. Please clarify.

Minor:

- I suggest avoiding to mix arousal and attention in the second paragraph of the introduction. As the authors describe in the discussion, attention (related to gain modulation of specific representations) fits better with the category-specific rather than the general process. It is better to focus this paragraph on arousal and the respective literature only.
- When talking about multivariate pattern analysis in the abstract, clarify that these are global topographies (rather than regional fine-grained patterns).
- Instead of "stimulus category" and "reported category", I suggest using "objective stimulus category" and "reported stimulus category" to avoid confusion regarding the differences across the various prediction approaches especially in the main text. The figures do a good job in this regard.
- Fig. 1A could include hypotheses with respect to F H O A categories in the specific model. I assume the brain state depicted as pink should result in "A" and the one depicted as purple should result in "H"?
- There is an interesting set of studies on general, presumably arousal-related brain state fluctuations in fMRI. A discussion of these (e.g. Chang et al. PNAS 2016 <https://doi.org/10.1073/pnas.1520613113>) would make this paper timelier and more integrated into the current debate.

We thank all three reviewers for their thoughtful and constructive comments. Following the reviewers' suggestions, we have made the following changes to figures:

- New Figure 1D illustrates how trials are grouped into Hits, Misses, False Alarms and Correct Rejections for SDT analyses of recognition behavior;
- New Supplementary Figure 1B shows the categorization task confusion matrices.
- New Supplementary Figure 1C shows trial-history dependence in recognition behavior.
- Supplementary Figure 2A now additionally shows activation pattern map for the General model in scrambled-image trials.
- The previous Figure 3 is now expanded into Figures 3 and 4, with the following additions:
 - We expanded the illustration showing potential influences of spontaneous activity on object recognition (i.e., detection; Fig. 3B) and category discrimination (Fig. 3C).
 - Figure 4 now contains two additional panels (E-F) showing spontaneous activity's influence on subjects' category discrimination of real and scrambled images.

In line with the reviewers' suggestions, we have also thoroughly edited the text to clarify methods and introduction, explain the newly added results, and include additional discussion points. Below we address each of the reviewers' comments in detail.

Reviewer #1:

This manuscript reports the results of one MEG experiment (N=24) investigating the influence of pre-stimulus activity on perceptual decision making in the visual domain. Low-contrast images from four categories (faces, animals, houses, objects, and their scrambled counterparts) were presented for 66.67ms, and participants were instructed to a) report the image category, and b) rate their subjective "recognition experience" (yes or no, see below). The authors used the recorded MEG and behavioral data to test two models: a general model (pre-stimulus brain states "influence recognition of a forthcoming stimulus regardless of the stimulus content", l.48), and a specific model (pre-stimulus brain states "facilitate recognition in a category-specific manner", l.50). Using multivariate pattern analysis (MEG) in combination with SDT analysis (behavior), the authors find evidence in support of both models, and report that the two processes were doubly dissociable: "the general process shifted the criterion of recognition but had no effect on sensitivity, whereas the specific process enhanced sensitivity with no effect on criterion" (l.10). The authors conclude that their findings reveal an "unknown dual role of spontaneous neural activity in object recognition and provide a new framework for future research" (l.331). While the findings are potentially relevant and interesting, a couple of major points need to be addressed for a better understanding of these complex data.

Major points

1) I tend to disagree with one of the authors' concluding statements: "thus, sensory inputs merely bias intrinsic brain dynamics rather than acting as agents essential for perception" (l.330). The observation that even scrambled images (which only contain low-level features of the four image categories) shown for 66.67ms could still be successfully categorized above chance level (51.9%, l. 94) seems to be in conflict with that statement. If I am not mistaken, this observation rather shows that even weak sensory inputs clearly dominated the reported percepts.

We agree with the reviewer's statement - even weak sensory inputs can significantly influence perceptual behavior. We did not intend to claim otherwise and therefore we have now edited the last paragraph of the discussion section to avoid such confusion (lines 398 - 403, page 14, last paragraph):

“Much is known about how the brain represents visual objects. In natural settings, however, object recognition does not rely on sensory input alone, but is regulated by ongoing brain dynamics to a large extent. For example, a percept of an object can emerge in the complete absence of sensory stimuli (as in dreams and hallucinations) and, on the other hand, salient stimuli can go unnoticed (e.g., during inattentive blindness). Our findings reveal a heretofore unknown dual role of spontaneous neural activity and shed light on how the brain implements object recognition beyond mere extraction of sensory features. [...]”

Furthermore, following the reviewer’s observation we additionally investigated how the two spontaneous processes, General and Specific, influence categorization of scrambled images (Fig. 4E-F; lines 274 – 278, page 11, 1st paragraph):

“[...] The percentage of scrambled images categorized correctly was $40.39 \pm 2.8\%$ and $36.49 \pm 2.1\%$ according to the General model and $44.71 \pm 3.77\%$ and 36.1 ± 1.64 according to the Specific model in high and low probability trial groups. Notably, only the neural process acting according to the Specific model significantly biased the category discrimination in scrambled images (Fig. 4F, $p = 3.45 \times 10^{-2}$ for Specific, $p = 0.14$ for General, $N=24$) [...]”

This result indicates that, while the reviewer is correct that even weak signal dominates the reported percepts, these percepts still can be biased by spontaneous brain activity.

2) With respect to the above-chance categorization performance (51.9%) for scrambled images, the authors write: “...suggesting that low-level image features ... contribute to categorization behavior and potentially to ‘visual hallucination’ (i.e., seeing a meaningful object out of meaningless patterns)” (l.97). I do not think that the above-chance categorization performance suggests that participants were “hallucinating” when confronted with the scrambled images. Rather, participants simply followed the instructions, as they were instructed to report the image category, and to guess it if necessary (l.66). So they used every low-level visual feature available to perform the categorization task. It remains unclear, however, how “vividly” they perceived the reported image category (i.e., the quality of their perception remains unknown). Therefore, I do not think that their performance can be compared to visual hallucinations (see also l.129). Furthermore, one potential weakness of the current paradigm seems to be that participants were not informed about the scrambled condition at all (l.515). It is not mentioned why this choice was made, and what its implications for participants’ behavior might have been.

We completely agree with the reviewer in that above-chance categorization performance does *not* suggest a “hallucination”. We acknowledge that the text in the previous version of the manuscript was confusing. We meant that the significant false alarms rate (in the responses to the recognition question, not categorization question) may imply such an interpretation. We now add this explanation early in the text (lines 79 - 87, page 2, last paragraph) and have removed all references to “hallucination” (such as l.97, l.129 mentioned by the reviewer):

“The recognition rate of scrambled images was $17.2 \pm 3.3\%$ [...] significantly above 0 (Wilcoxon signed-rank test, $p = 4.0 \times 10^{-5}$). The subjects were instructed to reply positively to a question about their recognition experience only if they could recognize an object in the stimulus presented; therefore, a report of recognition of a scrambled image constitutes a “False Alarm”. There are two possible interpretations of the significant False Alarm rate: (1) the subject perceived an object even though no object was present, and (2) the subject did not actually perceive an object but pressed a wrong report button, either by accident or because they forgot what they saw. The report of recognition experience is inherently subjective; therefore, we cannot discern between these two possibilities.”

We also clarified the reasons for the use of scrambled images (page 2, 3d paragraph, lines 70 - 72):

“Because scrambled images did not include an object stimulus, they were used as “catch trials” to determine the subjects’ baseline tendency to give positive responses to a question about their recognition experience.”

Not informing subjects of the catch trial condition could potentially induce a shift of criterion in the liberal direction. However, consistent with previous studies, we found a conservative criterion in the recognition responses (page 3, 2nd paragraph, lines 100 - 102):

“The subjects’ recognition reports had [...] a positive (i.e., conservative) decision criterion ($c = 0.66 \pm 0.11$, Wilcoxon signed-rank test, $p = 10^{-4}$), in line with previous studies (Iemi et al., 2017; Li et al., 2014; Limbach and Corballis, 2016; Wyart and Tallon-Baudry, 2008).”

3) Related to my previous point, the authors do not address any potential weaknesses of their design, or alternative models that could explain their data. Overall, there seems to be too little room for doubt and nuance in this manuscript.

It is not clear to us what kind of alternative models the reviewer has in mind. We tested two models describing the influence of spontaneous neural activity on object recognition behavior and, at least in the context of “content-specificity”, there is no third alternative. We designed the experiment and data analyses with the intention to avoid ambiguity; therefore, the weaknesses of the study that we know of do not compromise the central findings and conclusions, but rather constitute a bottleneck for further investigation and are discussed in 2nd paragraph, page 14 (lines 379 - 386):

“The current study does not provide substantial insights into the anatomical underpinnings of the general and specific spontaneous neural processes. [...] Future studies employing intracranial recordings or fMRI should help with providing anatomical insight [...]”

4) It is not mentioned how the sample size (N=24; for some analysis N=15, 22) was determined.

We now added this information in methods section (methods, page 15, lines 413 – 414):

“We chose a sample size similar to those used in recent published MEG studies on perception and cognition conducted with healthy human volunteers.”

We note that this is not a clinical-trial study (per current NIH definition), where the investigators typically have an a priori estimate of effect size and a power analysis is run before the study is conducted. Our sample size is in line with current typical perceptual or cognitive studies using MEG in healthy human volunteers, and all of our analyses employed rigorous statistical tests such as cross-validation and nonparametric permutation test.

We have also clarified the inclusion criteria for MVPA analyses (methods, page 18 last paragraph, lines 560 - 565):

*“In analyses that involved calculating AUROC for decoding of the recognition report in scrambled images (Fig. 2A, 2nd to 4th bars), only subjects that reported at least 5 scrambled images as recognized were included (N = 15). **This minimal number of trials was required in order to fit the model (i.e., “training trial set”) and to derive a meaningful interpretation of the model’s discriminability from a separate set of trials (i.e., “testing trial set”).** All 24 subjects were included in all other decoding analyses, which did not rely on scrambled images (Figs. 2B-E, 4).”*

We now include all subjects in the SDT analysis (N=24). Two subjects were originally excluded from the SDT analysis due to having zero false alarm rate (which leads to infinite values of SDT metrics). The exclusion was no longer necessary after we implemented the Macmillan & Kaplan correction method (line 578, page 19):

“We implemented Macmillan & Kaplan correction (Macmillan and Kaplan, 1985) of FAR = 0 and HR = 1: ...”

5) Terminology. The authors write: “Object recognition can happen in the complete absence of sensory stimuli (as in dreams and hallucinations)” (l.329). I would argue that “recognition” and “perception” cannot be used interchangeably, as it is done here. Similarly, the authors use the terms “recognition”, “subjective recognition”, “recognition rate”, and “recognized trials” with respect to their second task, in which participants “were instructed to report their recognition experience” (l.511-516). The task is more like a subjective visibility rating (e.g., perceptual awareness scale), and it should be consistently framed and labelled as such (e.g., “high” and “low” visibility, or “yes” and “no” trials, as opposed to “recognized” and “unrecognized” trials). The current terminology is somewhat misleading.

We edited the text in line 400 of the revised manuscript (line 329 originally) according to the reviewer’s suggestion:

“[...] a percept of an object can emerge in the complete absence of sensory stimuli (as in dreams and hallucinations)”

Consistent with the reviewer’s suggestion, we use “Y” and “N” abbreviation for “yes” and “no” trials in all figures. We have also added an illustration in Figure 1D about the definition of Hit, Miss, False Alarm and Correct Rejection trials, and clarified in line 77:

*“Participants reported $44.9 \pm 3.5\%$ (mean \pm s.e.m., $N = 24$) of real images as **recognized (i.e., % of “yes” reports)**”*

In threshold-level visual detection literature, the trials are typically referred to as “seen” and “unseen”. In threshold-level object recognition literature, such as studies using object images and backward masking, the terminology is conveniently adapted to “recognized” and “unrecognized” (Bar et al., 2001; Grill-Spector et al., 2000), which we follow in text for the sake of consistency and readability.

In particular, our paradigm here differs from previously adopted threshold-level perception paradigms that emphasize visibility of low-level features (e.g. Wyart and Tallon-Baudry, 2008; Li et al. 2014), but is more in line with threshold-level object recognition studies that investigate subjective perception of high-level meaningful content (e.g. Bar et al., 2001; Grill-Spector et al., 2000). We have now clarified this point (page 2, paragraph 2, lines 55 - 61):

“In order to calculate the subjective recognition rate, the subjects were instructed to report whether they see an object (“yes”/“no”), such that even if the object appears unclear or noisy they should respond “yes”, but if they see nothing or see only low-level features, such as lines or cloud-like abstract patterns, they should respond “no”. The present paradigm is analogous to threshold-level visual detection tasks using simple low-level stimuli (van Dijk et al., 2008; Li et al., 2014; Wyart and Tallon-Baudry, 2008), but with important differences in stimulus type (Gabor patches vs. objects) and the definition of threshold (stimulus visibility vs. object recognition).”

6) More details about the multivariate MEG analysis would be helpful and enable readers to fully understand the reported results. What MEG data exactly enters this analysis? How does this

multivariate analysis relate to previous EEG/MEG studies on pre-stimulus activity and its influence on perception?

We now clarify the methods in lines 549 - 554 in “Multivariate pattern analysis” section:

“Pre-stimulus brain state, X , was calculated using MEG data recorded from M sensors and averaged across time in a 2-second window before stimulus onset (see MEG data preprocessing above for more details) and, for every experimental trial, X constituted an M -dimensional vector. In this scenario, data from each MEG sensor constitutes a model feature, and β_c is an M -dimensional vector of weights. All sensors were used to fit the models except the analyses in Supplemental Figure 2B and 2D where separate groups of sensors were used to fit and test each model.”

Additional data pre-processing steps can be found in the methods section “MEG data preprocessing”.

We discuss how the present findings relate to previous EEG/MEG studies on pre-stimulus activity in second paragraph of discussion.

7) The results are analysed within a SDT framework, following “standard SDT analysis” (l.622). Low-contrast images from four categories (faces, animals, houses, objects; “real” images) and their scrambled counterparts were presented. These “real” and “scrambled” images were presented in random order, and they make up the total stimulus set. Thus, the participants’ behavior in the two tasks (image categorization, and subjective rating of “recognition experience”) is linked tightly to this stimulus set in its entirety. SDT analyses on subsets of these stimuli (e.g., “analyses that involved scrambled images”, l.633) are therefore problematic at least, if not invalid (see lines 633-635).

The reviewer is correct in that the SDT analyses require both real and scrambled images. This is exactly what we did. We realize that the phrasing in l.633 of the original manuscript was ambiguous. We now add an illustration in figure 1D showing that “Hits” were defined by real-image trials and “False Alarms” by scrambled-image trials. The new illustration in figure 3B further clarifies that both real and scrambled image trials together define criterion and sensitivity. We explain such depiction in text (page 8, last paragraph, lines 233 - 235 in revised text):

“In SDT framework, the task of recognition experience report (“yes”/“no”) can be interpreted as a signal detection task, where “signal” refers to an object stimulus - present in real images and absent in scrambled images.”

This is also clarified in methods section “Signal detection theory (SDT) analysis” in lines 571 - 577 of the revised manuscript.

The method of splitting the trials into groups according to an independent variable for calculating shifts in criterion and changes in sensitivity that correlate with that variable has been previously reported (e.g., lemi et al., 2017; Urai et al., 2017).

8) As this is a categorization task with four categories, confusion matrices (for real and scrambled trials) would be informative, and could easily be added to the supplement. For example, it might turn out that faces were often correctly categorized, while the other categories were harder to categorize for participants.

The confusion matrices for the categorization task, split by real and scrambled image trials, are now added as Supplemental figure 1B.

The reviewer is correct in that differences in categorization accuracy across categories can be observed, especially in scrambled-image trials. Because the only analysis in the original manuscript that used the categorization reports was presented in fig 2E, we anticipate that the reviewer may be

interested in how such analysis was statistically validated, given the imbalance in category reports. In order to test the models in fig 2E we performed a label permutation test – that is, the actual reported categories were shuffled across trials (i.e., preserving the imbalance) and the model was tested 1000 times using such permuted labels for each subject. Furthermore, the area under the receiver-operator curve (AUROC) metric we use to evaluate the performance of the model for each subject is insensitive to class imbalance. These details are given in the methods section “Multivariate pattern analysis” paragraph 2 with additional clarifications.

9) I did not understand how the trials were split into two groups for each process (I.205). Was this done separately for each participant? Did this procedure result in the same number of trials in each group?

The split was indeed done for each participant (e.g., in Figures 3B-E (original) or Figure 4A-D (revised) each data point represents a participant and we performed a paired test between groups).

The procedure (explained in detail by the schematic in Supplementary Fig 3) did not consider the number of trials: for each participant and for each trial, the model’s prediction score (e.g., probability of recognition) was calculated and the trial was placed in the high or low probability group based on this score. Thus, the number of trials was not necessarily the same between groups for each participant.

However, this inequality in trial number between groups does not affect our analysis or interpretation, because the statistical significance of the difference across groups was assessed by nonparametric paired tests (Wilcoxon signed-rank), where the variability (across subjects) of *the difference between groups* is considered (unlike variability of each group in two-sample tests). In addition, unlike decoding accuracy, SDT metrics such as d' and c are not systematically biased by trial count (i.e., lower trial counts produce noisier d' and c measurements, but not systematically lower or higher measurements).

10) The results are primarily described in terms of “significant” ($p < .05$) and “not significant” ($p > .05$). It would be informative to learn more about the size of the effects, at least some descriptive stats could be added to the main text (e.g., when writing “ d' was influenced by pre-stimulus activity patterns”, or “ d' was not influenced by...”, the respective mean d' values could be added, in addition to the figures). Similarly, the important double dissociation between shifts of criterion and enhancement of perceptual sensitivity (lines 221-232) relies on “significant” and “non-significant” p-values alone, while it seems more appropriate to directly test the interaction (i.e., “model x hi/low” interaction, separately for d' and c , Figure 3).

We have now added the effect sizes in text throughout the results section.

Double dissociation (commonly defined as A having an effect on X but not Y, while B having an effect on Y but not X; see left panel below) does not necessitate a significant interaction. In fact, a significant interaction could mean a lack of double dissociation (e.g., right panel below). It is interesting to find a double dissociation because it points to the two ongoing brain processes having separate downstream neural mechanisms.

Reviewer #2:

Review of: A dual role of spontaneous neural activity in visual object recognition

The present study investigated brain states that facilitate visual perception. The authors demonstrated, using magnetoencephalography (MEG) recordings, that certain brain-states enable better recognition of visual stimuli, presumably by lowering criterion threshold (i.e. higher propensity of a 'yes' answer), and furthermore that specific brain-states facilitate visual recognition of stimuli related to specific categories (man-made objects, animals, houses and faces).

The notion that the brain has intrinsic states that facilitate (or hamper) visual recognition is interesting and worthy of investigation. The paper is well-written and the study seems rigorous and well-executed.

We thank the reviewer for the interest and the thoughtful comments.

Several comments that should be addressed are listed below for consideration:

1. Does the brain state that facilitate recognition of one category similar to the one facilitating perception of a stimuli from another category? Given that no source localization method seems to have been undertaken here, a precise answer for this question might not be possible, but the topoplots presented in supplemental Fig 2 suggest that a similar pattern of activation preceded all of the four categories. How can the authors explain a similar activation for diverse categories in the specific model proposed?

Because the specific model can predict the stimulus category better than chance (Figure 2C) – the MEG activity patterns preceding the recognized images from different categories are, at least partly, different (otherwise the model would fail). If so, why are the patterns in supplemental Figure 2 similar?

First, we would like to clarify that the activation patterns that we present in supplemental Figure 2 are *not* the topographies of actual brain activity preceding each stimulus category. Instead, these are the patterns derived from the weights of a decoder that distinguishes each category from other categories. In a simple case of only two categories, we need only one set of weights – that is, exactly the same activation pattern map for two different categories. In the case of four categories, we in fact estimate the model weights for each category (discriminating that category from others) and the decision of the classifier is based on the distance of each trial's data point from each hyperplane defined by these weights. Consider a three-category classification example (adopted from sklearn):

In this example, the two dimensions depicted (“features”) are highly informative for the classifier but have very similar weights for blue and yellow classes (indicated by blue and yellow dashed lines). Note that the means of the data points (blue and yellow) are very different despite the very similar weights (our decoders had 272 features corresponding to 272 MEG sensors, and the decoder was trained for each subject). It is not possible to tell from looking at the weights alone how well the model will perform or how different the activity topographies themselves are. For this reason, we provided rigorous leave-one-out cross-validated performance metrics, statistically assessed using label permutation tests.

Second, the weights are determined by cross-information maximization algorithm which can result in a non-zero weight of a feature that is in fact representing noise. Such “noise-suppressive” weights can be confusing when attempting to interpret the most informative features. This problem is addressed in great detail in (Haufe et al., 2014), who developed a procedure of transforming model weights into activation patterns by multiplying by the data covariance matrix. We utilized their technique to calculate the activation patterns presented in Supplementary Figure 2. We acknowledge that the details of the analysis in supplemental Figure 2 were missing and we have now clarified them in the methods section: “Multivariate Pattern Analysis” page 18, lines 555 - 559.

In sum, the gross similarity of the activation pattern maps associated with different categories as shown in Supplementary Figure 2 suggests that similar sensors (indicated by blue and red) contribute the most to the classification of each category against others. These sensors overly occipitotemporal areas most prominently, consistent with the brain-region-specific decoding results shown in panel F. The fact that the decoder is able to distinguish across categories is because the actual pre-stimulus MEG activity pattern differs between recognized trials where real images from different categories are presented.

We would also like to point out a minor change in supplemental Figure 2A and C – the activation patterns presented in the originally submitted manuscript corresponded to an earlier version of the software and used a slightly different fitting algorithm. We recomputed the patterns to be consistent with all other analyses presented (the difference in patterns is hardly noticeable but we wanted to point this out to avoid any confusion).

2. Anticipations can make one category subsequently more or less recognizable, but expectations are not a brain “state.” These are more different types of prepared-nesses that might stem from previous trials as well as the perceptual history with which the participants arrived to the experiment.

We agree with the reviewer. First, we have included a discussion paragraph to address these points (page 14, 1st paragraph, lines 371 - 378):

“It is possible that the spontaneous neural processes reported herein partially reflect spontaneous fluctuations in cognitive processes such as attention and expectation. While in the present study we did not explicitly manipulate these cognitive processes, spontaneous fluctuations in neural activity are suggested to correlate with spontaneous fluctuations in cognition (Summerfield and de Lange, 2014). Furthermore, top-down cognitive processes influence object recognition (Baldauf and Desimone, 2014; O’Craven et al., 1999). It is also possible that bottom-up visual signals simply encounter the ongoing

neural state of the local networks along the visual processing hierarchy, which in turn affects processing of these signals. These possibilities of the relationship between spontaneous neural processes and top-down/bottom-up stimulus processing deserve future investigation.”

Furthermore, in an attempt to shed more light on this question, we examined trial history effects in behavior. We found that when a trial was reported as recognized, the following trial was more likely to be recognized as well. However, a previous trial of the same objective stimulus category did not influence subjective recognition rate. These results are reported in Supplementary Figure 1 and Results section “Paradigm and behavior” (page 3, lines 88 - 96).

We also examined the relationship between predictions made by the two models and subjective recognition reports on the previous or the following trial, but did not find any significant effects there. This analysis is now reported in a new Results section: “Trial-history in General and Specific processes” (page 11, lines 284 - 297).

We agree with the reviewer that the question of individual perceptual history (both within and outside the experimental session) is interesting and worthy of investigation, but our experiment was not specifically designed to address these questions – e.g., the long pre-stimulus intervals adopted here to better investigate spontaneous activity may have weakened potential trial-to-trial effects.

3. Spontaneous activation might result from persistence of some sort (especially because presentations did not use backward masking) and from top-down influences. A thoughtful discussion of top-down vs bottom-up influences on pre-stimulus “spontaneous” activity would be important and relevant.

We thank the reviewer for the suggestion to expand the discussion on top-down vs. bottom-up processing. This is now addressed in page 14, 1st paragraph, quoted above in the reply to point #2.

We note that persistence of stimulus information influencing pre-stimulus activity of the ensuing trial is highly unlikely in our experiment, due to the long trial length (~10 sec), the long pre-stimulus interval (3-6 sec), and the intervening multiple response screens in every trial (see Fig. 1B).

The null results mentioned above in response to the reviewer’s point #2 about a lack of trial-history effects in the General or Specific process support this conclusion (described in detail in Results section “Trial-history in General and Specific processes”, page 11). For example, the objective (i.e., actually presented) stimulus category in the previous trial had no influence on the category prediction made by the Specific model based on pre-stimulus activity in the current trial, as reported in lines 289 - 291:

“[...] the model-predicted category matched the objective stimulus category presented on the previous trial at a rate of 25.7 ± 0.5 %, not higher than the chance level of 25% (Wilcoxon signed-rank test, $p = 0.13$).”

4. Supplemental Fig 2 is a topoplot portraying the topography of activation prior to presentation of real-image trials. However, it is stated that this brain-state could also predict scrambled trials (as shown in Fig 2A). Was the pattern of activation in this general model similar in both instances?

The activation pattern map for scrambled images is now added to supplementary Figure 2 - the activation patterns are similar between real- and scrambled-image trials (but see our reply to the reviewer’s point #1 about the interpretation of activation pattern maps, and why they are not trivially “topography of activation”). We note that the cross-decoding analysis tested for statistical significance with label permutation test provides more robust evidence of the General model’s generalization across image types than activation pattern similarity.

5. The claim about doubly dissociable influences on object recognition (i.e. criterion and sensitivity) sounds a bit circular. The general and specific models were initially constructed by predicting correct recognition (a 'yes' answer for the general model or selection of a correct category in the specific model). In that case, isn't the association between the general model and criterion, and between the specific model and sensitivity, a by-product of this initial analysis?

Please allow us to clarify that the SDT metrics were derived from subjective recognition reports alone ('yes'/no'; 2nd question in Fig. 1B) in real and scrambled images ('yes' corresponds to Hits and False Alarms, respectively) for both General and Specific models. Correct/incorrect category reports (question 1) were *not* used in the SDT analysis in the original manuscript (but added in the revised version, please see below). In addition, the trials selected by the Specific model into the high probability group were those where the category predicted by the model matched the *objective* (i.e., actually presented) stimulus category. In other words, the *model* predicted the upcoming stimulus category correctly, regardless of what the subject responded. To avoid any confusion, we now specify "*objective stimulus category*" vs "*reported stimulus category*" in the revised text.

Thus, the *double dissociation* analysis outcome is not a by-product of model construction. We now clarify this in text (page 8, last paragraph, lines 231 - 235) and in illustration (Fig. 3):

"The results so far indicate that both ongoing brain processes, General and Specific, influence subjective recognition, but do not inform on whether such effects stem from a shift in criterion, a change in sensitivity, or both. In SDT framework, recognition experience report ("yes"/"no") can be interpreted as a signal detection task, where "signal" refers to object information – present in real images and absent in scrambled images.[...]"

Moreover, both General and Specific models were trained using real-image trials only and how each model's prediction biases perceptual behavior in scrambled-image trials (which is crucial for generating SDT metrics) is not a trivial outcome. In addition, the models were cross-validated within real trials (lines 229 – 230, page 8):

" [...]all models for this analysis were trained using real-image trials only, when the probabilities were predicted for scrambled-image trials and real-image trials through cross-validation."

Lastly, we anticipate that the reviewer may be interested in knowing whether the General and Specific spontaneous processes also influence subjects' categorization behavior (1st question in Fig. 1B, new illustration in Fig. 3C). We performed such an analysis separately for real and scrambled images and the results are now presented in figure 4E-F and in text (1st paragraph on page 11). Briefly, we found that while both the General and Specific processes influenced categorization accuracy in real-image trials, only the Specific process influenced categorization accuracy in scrambled-image trials.

Some technical issues:

6. Given that eye-tracking was administered in this study, it is advisable to check for times where participants did not look at the screen and thus did not recognize the stimuli (i.e. instances in which the brain state could not predict visual perception simply because participants weren't looking).

Trials with blinks have been removed, see Methods section "MEG data preprocessing". The subjects were instructed and reminded between blocks to fixate on the center crosshair and to avoid blinking to the best they can until after stimulus arrival.

We included eye-tracking in this study primarily to monitor pupil size. Unfortunately, we did not perform proper eye-position calibration at each experiment block due to experiment duration constraints and therefore cannot conduct a rigorous analysis of fixation behavior at the single-trial level. Nonetheless, in order to check that the subjects did generally follow the instruction we used the uncalibrated eye position and performed the following control analysis. First, to allow comparing across blocks and subjects, we removed the mean of eye position (in x and y coordinates) and divided by standard deviation from each block. Next, for each trial we calculated the mode of eye position in the time window from 100 ms before stimulus onset to 100 ms after, which provided a rough estimate of the fixation position during stimulus presentation. Last, we calculated the heatmap of fixation positions across trials for each subject and plotted the average across subjects. We also performed the same analysis for eye positions recorded during the response prompts, where a question and response mapping appeared on the screen (Fig. 1B, 'report'). Results are presented below.

Eye position during stimulus onset:

Eye position during report:

These results show that subjects generally followed the instruction and centered their gaze before/during the stimulus appearance in comparison to the text screen appearance. However, without calibration we cannot confidently reject single trials based on this metric. We opted to not include this analysis in the paper due to lack of sufficient rigor.

7. What is the reason for a non-parametric testing across the analyses (instead of parametric tests)?

We now better explain in the methods section how we made the choice (page 19 bottom):

“Potential group-level differences between paired groups of bounded variables (AUROC, %) were tested using nonparametric two-tailed Wilcoxon signed-rank tests.”

Following the reviewer’s comment, we now use paired t-tests (parametric test) to assess differences in sensitivity and criterion, which follow normal distribution:

“One-sample Student t-test was used to assess differences in criterion and sensitivity”

We used label permutation tests to assess the significance of predictions made by the multivariate models for a different reason: the theoretical chance level of supervised models (e.g., 0.5 for 2 classes) is achieved only with infinite number of trials (Combrisson and Jerbi, 2015) - therefore, label permutation test to derive data-driven chance level is a much more rigorous way of testing classifier performance (page 18, lines 539 - 544):

“... we repeated the analysis 1000 times with labels shuffled across trials in the training set and re-computed the AUROC value for each permutation - this procedure was used to estimate the data-driven chance-level distribution of AUROC values(Combrisson and Jerbi, 2015)”

8. A scale indicating activation values is missing in supplemental Fig 2.

We have added the scale now.

9. Under the methods section it is stated that one participant dropped out (thus N=23), yet in the results N=24 is indicated.

We thank the reviewer for catching this mistake. The data analyses indeed included 24 subjects but the correct number of subjects enrolled in the study is 25:

“All participants (N=25, 15 females, mean age 26, range 22 to 34) provided written informed consent.”

Reviewer #3:

This work represents an important advance in the understanding of the functional role of spontaneous brain activity. Similar to several other works, it studies the impact of pre-stimulus activity on subsequent perception of threshold stimuli using a prediction approach, however with the important novelty of separating fluctuations in a more general “content-unspecific” neural process and a category-specific neural process. As the current interest in spontaneous activity is very high (apparent e.g. in the countless resting state studies), this MEG study will be of broad interest. A few specific questions / suggestions follow.

We thank the reviewer for the enthusiasm and the helpful comments.

The machine learning predictions are based on MEG sensor-level signals averaged over a 2s pre-stimulus period. In the discussion section the authors acknowledge the lack of topographical information due to methodological limitations. I am puzzled about what information is being harnessed by the model for the category-specific predictions? My assumption would be that brain activity dissociating upcoming perception across visual object categories would be localized in the ventral visual stream, an area that is hard to detect in MEG signals even if source-localization was possible.

First, please allow us to point out that information related to visual object processing in the ventral stream is readily available in MEG signals as was demonstrated using source localization (Bar et al., 2006) and in numerous recent publications that decode object category from whole-head MEG signals (e.g. Carlson et al., 2013; Ritchie et al., 2015; Vida et al., 2017). In addition, recent studies using cross-modal representational similarity analysis (RSA) have confirmed that such object information decoded from whole-head MEG correlates with information originating from ventral temporal cortex in human fMRI (Cichy et al., 2016; Hebart et al., 2018) and IT in monkey (Cichy et al., 2014).

Second, we agree with the reviewer and have the same hypothesis in mind. We have now added this point to discussion (lines 391 – 393, page 14):

“[...] it remains to be tested whether the specific ongoing process is supported by brain areas specialized in object processing, such as the ventral temporal cortex (VTC). [...]”

Following a related line of thought, I am wondering whether the topographies for brain states preceding each of the four object categories provided in Supplementary Fig. 2C have similarity to the post-

stimulus states evoked by each of the respective categories? The authors discuss that the spontaneous (pre-stimulus) category-specific activity patterns could represent samples or knowledge about typical objects. I interpret this to mean that the patterns that facilitate perception of a certain category can be viewed as some degree of reactivation of neural patterns representing that category. Thus, they should have similarity to the post-stimulus evoked state of that category. In addition to a direct quantitative comparison of pre- and post-stimulus topographies, it would be interesting to see if training a model on the post-stimulus activity of the four categories could predict reported category when tested on pre-stimulus activity.

Unfortunately, non-source-localized MEG data do not allow a definite test for the reactivation hypothesis. This is because MEG topographic maps reflect not only magnitude, but also a **direction of brain electrical dipoles**. Thus, it is common to observe a reversal of pre- and post-stimulus activity patterns in MEG. A post-stimulus activity MEG topography at the whole-brain level is strongly dominated by the visual evoked activity and may mask reactivation at the local-circuit level (e.g., within the ventral temporal cortex).

With the above caveat in mind, we nevertheless performed the analysis suggested by the reviewer. We trained the model (same as Figure 2C, first bar) at a given pre- or post-stimulus time point to predict the objective stimulus category and tested it on all other time points in the trial (only recognized real-image trials were used). For this analysis we used 250-ms non-overlapping temporal windows. This analysis was implemented using the MNE decoding toolbox temporal generalization with 4-fold cross-validation, the results are quantified with F1 score (as 4-category auROC metric was not available in the toolbox).

We did not observe significant cross-decoding between pre- and post-stimulus time points (upper left and lower right quadrant). However, this does not mean that the ‘reactivation’ hypothesis as suggested by the reviewer can be rejected based on these findings. To definitively test this hypothesis, future investigation using techniques with higher spatial resolution (such as intracranial EEG or fMRI) is needed. We now include the ‘reactivation’ hypothesis as a direction for future research in Discussion (page 14, lines 393 - 396):

“Recent studies demonstrated the contribution of VTC to object category decoding from whole-brain MEG signals by employing novel computational approaches to combine MEG and fMRI data (Cichy et al., 2014; Flounders et al., 2019). Thus, reactivation of stimulus-triggered patterns in spontaneous VTC activity may underlie the specific ongoing process uncovered herein – a hypothesis that warrants future investigation.”

Regarding the Signal Detection Theory (SDT) application, I understand that both criterion c and sensitivity d' were estimated using hit and false alarm rates with respect to the second question, i.e. the question whether the subject recognized the object. This is in contrast to the approach of the rest of the paper up to this point, in which the general (category-insensitive) process was targeted using the 2nd question, while the category-specific process was targeted using the first question, i.e. which of four categories the subject saw. If I understand correctly, this SDT approach thus assumes that reporting “Yes” to the recognition question (2nd question) on a scrambled image is considered a false alarm, but the behavioral results showed that subjects were actually successful in using low-level features of the

scrambled images to correctly “recognize” the category far above chance. This implies that answering “Yes” on a scrambled trial is not actually a false alarm. Please clarify.

Please allow us to clarify that the category-specific model was not fit to predict subjective category report (1st question), but using the *objective* stimulus category (Figure 2C).

To address the reviewer’s question, it is important to point out that scrambled images, by construction (phase scrambling), do not contain an object. However, the reviewer is correct that low-level features of scrambled images do contain category-specific information allowing for better-than-chance categorization and it is possible to investigate how spontaneous process influences subjects’ categorization accuracy in scrambled images (a new analysis was performed to address this, see below). We now explain that our task has two dimensions: object detection (“yes”/“no”) and category discrimination (“F”/“H”/“O”/“A”) and we now add the analysis of categorization performance (see revised Figure 3C for illustration and Figure 4E-F for results). Below is a detailed list of all the changes we made following the reviewer’s comment:

First, we have clarified that subjects were instructed to report “yes” to a question about recognition experience only if they saw an object and to report “no” otherwise (page 2, last paragraph, lines 81 - 83):

“The subjects were instructed to reply positively to a question about their recognition experience only if they could detect an object in the stimulus presented, therefore, a “yes” report of a scrambled image constitutes a “False Alarm” (Fig. 1D).”

and in addition (page 8, lines 233 - 235):

“In SDT framework, recognition experience report (“yes”/“no”) can be interpreted as a signal detection task, where “signal” refers to an object stimulus – present in real images and absent in scrambled images.”

Furthermore, we clarified that low-level stimulus features alone could contribute to above-chance categorization accuracy (page 3, 3d paragraph, lines 110 - 112):

“the categorization accuracy [in scrambled images] was significantly above chance in both recognized and unrecognized trials ($p = 1.2 \times 10^{-3}$ and $p = 7.6 \times 10^{-3}$, respectively), suggesting that low-level image features that are distinct between categories (Coggan et al., 2016) contribute to categorization behavior”

We have further added an illustration on how spontaneous processes may influence detection and discrimination task performance in Figure 3B-C.

Lastly, we have carried out a new analysis to assess whether and how the General and Specific spontaneous process influences subjects’ categorization behavior. The results of this analysis are shown in the newly added Fig. 4E-F and described in detail on page 11 (1st paragraph). Related to scrambled images, we found (lines 276 - 278):

“[...]only the neural process acting according to the Specific model significantly biased the category discrimination in scrambled images (Fig. 4F, $p = 3.45 \times 10^{-2}$ for Specific, $p = 0.14$ for General, $N=24$) [...]”

Minor:

- I suggest avoiding to mix arousal and attention in the second paragraph of the introduction. As the authors describe in the discussion, attention (related to gain modulation of specific representations) fits better with the category-specific rather than the general process. It is better to focus this paragraph on arousal and the respective literature only.

We agree with the reviewer and have now removed the reference to attention in the Introduction.

- When talking about multivariate pattern analysis in the abstract, clarify that these are global topographies (rather than regional fine-grained patterns).

Done. Abstract: “[...] *multivariate pattern analysis applied to whole-brain activity* [...]”

- Instead of “stimulus category” and “reported category”, I suggest using “objective stimulus category” and “reported stimulus category” to avoid confusion regarding the differences across the various prediction approaches especially in the main text. The figures do a good job in this regard.

We thank the reviewer for this suggestion - we have now revised the text accordingly throughout the results section.

- Fig. 1A could include hypotheses with respect to F H O A categories in the specific model. I assume the brain state depicted as pink should result in “A” and the one depicted as purple should result in “H”?

Does the reviewer mean resulting in “A” and “H” category report by the subject? If so, the reviewer is correct, the reports of successful subjective recognition correlate with high accuracy of reported stimulus category. However, we have found (see Fig. 2D-E) that pre-stimulus states influencing the recognition according to the specific model are not necessarily the same as the states influencing the reported stimulus category in unrecognized images; therefore, we are afraid such a depiction could be misleading.

- There is an interesting set of studies on general, presumably arousal-related brain state fluctuations in fMRI. A discussion of these (e.g. Chang et al. PNAS 2016 <https://doi.org/10.1073/pnas.1520613113>) would make this paper timelier and more integrated into the current debate.

We thank the reviewer for this suggestion. We now refer to this and other studies on resting-state fMRI correlates of arousal (Chang et al. 2016, Yellin et al 2015, Schneider et al, 2016, Liu et al. 2018) in discussion (page 14, paragraph 3, lines 389 - 391):

“...recent reports suggest that it is possible to track arousal fluctuations using resting-state fMRI signals in widespread cortical and subcortical regions (Chang et al., 2016; Liu et al., 2018; Schneider et al., 2016; Yellin et al., 2015). A direct influence of these arousal-related modulations on sensory processing remains to be shown.”

References

- Baldauf, D., and Desimone, R. (2014). Neural Mechanisms of Object-Based Attention. *Science* (80-.). 344, 424–427.
- Bar, M., Kassam, K.S., Ghuman, A.S., Boshyan, J., Schmid, A.M., Schmidt, A.M., Dale, A.M., Hämäläinen, M.S., Marinkovic, K., Schacter, D.L., et al. (2006). Top-down facilitation of visual recognition. *Proc. Natl. Acad. Sci. U. S. A.* 103, 449–454.
- Carlson, T., Tovar, D.A., Alink, A., and Kriegeskorte, N. (2013). Representational dynamics of object

vision: The first 1000 ms. *J. Vis.* 13, 1–1.

Chang, C., Leopold, D.A., Schölvinc, M.L., Mandelkow, H., Picchioni, D., Liu, X., Ye, F.Q., Turchi, J.N., and Duyn, J.H. (2016). Tracking brain arousal fluctuations with fMRI. *Proc. Natl. Acad. Sci.* 113, 4518–4523.

Cichy, R.M., Pantazis, D., and Oliva, A. (2014). Resolving human object recognition in space and time. *Nat. Neurosci.* 17, 455–462.

Cichy, R.M., Pantazis, D., and Oliva, A. (2016). Similarity-Based Fusion of MEG and fMRI Reveals Spatio-Temporal Dynamics in Human Cortex During Visual Object Recognition. *Cereb. Cortex* 26, 3563–3579.

Coggan, D.D., Liu, W., Baker, D.H., and Andrews, T.J. (2016). Category-selective patterns of neural response in the ventral visual pathway in the absence of categorical information. *Neuroimage* 135, 107–114.

Combrisson, E., and Jerbi, K. (2015). Exceeding chance level by chance: The caveat of theoretical chance levels in brain signal classification and statistical assessment of decoding accuracy. *J. Neurosci. Methods* 250, 126–136.

van Dijk, H., Schoffelen, J.-M.J.-M., Oostenveld, R., and Jensen, O. (2008). Prestimulus oscillatory activity in the alpha band predicts visual discrimination ability. *J. Neurosci.* 28, 1816–1823.

Flounders, M.W., González-García, C., Hardstone, R., and He, B.J. (2019). Neural dynamics of visual ambiguity resolution by perceptual prior. *Elife* 8.

Haufe, S., Meinecke, F., Görgen, K., Dähne, S., Haynes, J.-D., Blankertz, B., and Bießmann, F. (2014). On the interpretation of weight vectors of linear models in multivariate neuroimaging. *Neuroimage* 87, 96–110.

Hebart, M.N., Bankson, B.B., Harel, A., Baker, C.I., and Cichy, R.M. (2018). The representational dynamics of task and object processing in humans. *Elife* 7, e32816.

Iemi, L., Chaumon, M., Crouzet, S.M., and Busch, N.A. (2017). Spontaneous Neural Oscillations Bias Perception by Modulating Baseline Excitability. *J. Neurosci.* 37, 807–819.

Li, Q., Hill, Z., and He, B.J. (2014). Spatiotemporal Dissociation of Brain Activity Underlying Subjective Awareness, Objective Performance and Confidence. *J. Neurosci.* 34, 4382–4395.

Limbach, K., and Corballis, P.M. (2016). Prestimulus alpha power influences response criterion in a detection task. *Psychophysiology* 53, 1154–1164.

Liu, X., De Zwart, J.A., Schölvinc, M.L., Chang, C., Ye, F.Q., Leopold, D.A., and Duyn, J.H. (2018). Subcortical evidence for a contribution of arousal to fMRI studies of brain activity. *Nat. Commun.* 9, 395.

Macmillan, N.A., and Kaplan, H.L. (1985). Detection Theory Analysis of Group Data. Estimating Sensitivity From Average Hit and False-Alarm Rates. *Psychol. Bull.* 98, 185–199.

O’Craven, K.M., Downing, P.E., and Kanwisher, N. (1999). fMRI evidence for objects as the units of attentional selection. *Nature* 401, 584–587.

Ritchie, J.B., Tovar, D.A., and Carlson, T.A. (2015). Emerging Object Representations in the Visual System Predict Reaction Times for Categorization. *PLOS Comput. Biol.* 11, e1004316.

Schneider, M., Hathway, P., Leuchs, L., Sämann, P.G., Czisch, M., and Spormaker, V.I. (2016). Spontaneous pupil dilations during the resting state are associated with activation of the salience network. *Neuroimage* 139, 189–201.

Summerfield, C., and de Lange, F.P. (2014). Expectation in perceptual decision making: neural and computational mechanisms. *Nat. Rev. Neurosci.* 15, 1–12.

Urai, A.E., Braun, A., and Donner, T.H. (2017). Pupil-linked arousal is driven by decision uncertainty and alters serial choice bias. *Nat. Commun.* 8, 14637.

Vida, M.D., Nestor, A., Plaut, D.C., and Behrmann, M. (2017). Spatiotemporal dynamics of similarity-based neural representations of facial identity. *Proc. Natl. Acad. Sci. U. S. A.* 114, 388–393.

Wyart, V., and Tallon-Baudry, C. (2008). Neural Dissociation between Visual Awareness and Spatial Attention. *J. Neurosci.* 28, 2667–2679.

Yellin, D., Berkovich-Ohana, A., and Malach, R. (2015). Coupling between pupil fluctuations and

resting-state fMRI uncovers a slow build-up of antagonistic responses in the human cortex.
Neuroimage 106, 414–427.

REVIEWERS' COMMENTS:

Reviewer #1 (Remarks to the Author):

The authors have addressed my comments satisfactorily.

Reviewer #2 (Remarks to the Author):

the authors have worked hard and wise to address my comments and i have no further remarks. but i would add that the references are pretty selective and not really reflective of the true origins of many of the ideas discussed. and so it goes... in any event, i commend the authors for an interesting study well-done.

Reviewer #3 (Remarks to the Author):

The authors have addressed all my major questions and suggestions in a thorough and detailed response. Especially the clarification that the second answer of the subjects ("Yes/No" question) entailed the detection (presence) of an object rather than reporting the success/failure of recognizing its category was key for me to follow the SDT-dependent analyses. The authors have added "The subjects were instructed to reply positively to a question about their recognition experience only if they could detect an object in the stimulus presented, therefore, a "yes" report of a scrambled image constitutes a "False Alarm" (Fig. 1D)."

This makes me wonder why the authors call this a "recognition question" rather than a "detection question". Would it be a valid move (depending on how the question to the subjects was precisely formulated) to revise the nomenclature accordingly?

Regarding the response to my minor comments:

I continue to believe that Fig. 1A should be extended to visualize the comprehensive set of the authors' predictions. This figure is supposed to represent the study's hypotheses. It is incomplete since the authors test hypotheses regarding the impact of prestimulus activity on both subjective detection (Y/N answer; cf. hypothesis tests represented in Figs. 2a-c) as well as reported category (F/O/A/H answer; cf. hypothesis tests represented in Figs. 2d-e). On a similar note, for the new Fig.1D illustrating trial types, the outcomes of the recognition/detection question are incorporated but not those of the categorization question. I think it is extremely confusing to the reader when hypotheses are represented with respect to one behavioral outcome but tests on the other outcome are also crucial to assess the validity of the specific model.

The authors' responded that: "However, we have found (see Fig. 2D-E) that pre-stimulus states influencing the recognition according to the specific model are not necessarily the same as the states influencing the reported stimulus category in unrecognized images; therefore, we are afraid such a depiction could be misleading."

What the authors hypothesized regarding the impact of prestimulus brain states on categorization should not depend on what they found in their data. I understand this will be visually complicated but maybe illustrations along the lines of those in Fig. 3 could help?

The phrase "multivariate pattern analysis applied to whole-brain activity" in the abstract could more explicitly point to sensor-level topographies, e.g. "multivariate pattern analysis applied across all MEG sensors" or "applied to the MEG sensor-level activity pattern".

Point-by-point response to reviewers

Reviewer #1 (Remarks to the Author):

The authors have addressed my comments satisfactorily.

Thank you!

Reviewer #2 (Remarks to the Author):

the authors have worked hard and wise to address my comments and i have no further remarks. but i would add that the references are pretty selective and not really reflective of the true origins of many of the ideas discussed. and so it goes... in any event, i commend the authors for an interesting study well-done.

Thank you!

Reviewer #3 (Remarks to the Author):

The authors have addressed all my major questions and suggestions in a thorough and detailed response.

Thank you!

Especially the clarification that the second answer of the subjects (“Yes/No” question) entailed the detection (presence) of an object rather than reporting the success/failure of recognizing its category was key for me to follow the SDT-dependent analyses. The authors have added “The subjects were instructed to reply positively to a question about their recognition experience only if they could detect an object in the stimulus presented, therefore, a “yes” report of a scrambled image constitutes a “False Alarm” (Fig. 1D).”

This makes me wonder why the authors call this a “recognition question” rather than a “detection question”. Would it be a valid move (depending on how the question to the subjects was precisely formulated) to revise the nomenclature accordingly?

We defined object recognition (2nd sentence of introduction) as:

“experiencing a percept of an object from physical properties of the stimulus projected to the retina”

We instructed the subjects according to this definition. The subjects were required to report a “yes” if pixels on the screen looked like an object or part of an object, and to report a ‘no’ if the pixels on the screen did not form a meaningful object in their mind - i.e., we targeted subjective experience. The question on the screen was not about object presence but about this subjective experience. One might think that changing the question on the screen to “was there an object?” would not make a difference, but this assumption necessitates an empirical test we did not perform; therefore, changing the nomenclature would be misleading. While Signal Detection Theory (SDT) has proven useful for analyzing our data, for the sake of successful replication of our experiment we prefer to keep the “recognition question” terminology.

Regarding the response to my minor comments:

I continue to believe that Fig. 1A should be extended to visualize the comprehensive set of the authors’ predictions. This figure is supposed to represent the study’s hypotheses. It is incomplete since the authors test hypotheses regarding the impact of prestimulus activity on both subjective detection (Y/N answer; cf. hypothesis tests represented in Figs. 2a-c) as well as reported category (F/O/A/H answer; cf. hypothesis tests represented in Figs. 2d-e). On a similar note, for the new Fig.1D illustrating trial types, the outcomes of the recognition/detection question are incorporated but not those of the categorization question. I think it is extremely confusing to the reader when hypotheses are represented with respect to one behavioral outcome but tests on the other outcome are also crucial to assess the validity of the specific model.

The authors' responded that: "However, we have found (see Fig. 2D-E) that pre-stimulus states influencing the recognition according to the specific model are not necessarily the same as the states influencing the reported stimulus category in unrecognized images; therefore, we are afraid such a depiction could be misleading."

What the authors hypothesized regarding the impact of prestimulus brain states on categorization should not depend on what they found in their data. I understand this will be visually complicated but maybe illustrations along the lines of those in Fig. 3 could help?

We absolutely agree with the reviewer - the hypothesis illustration should not depend on data. However, we would like to draw the reviewer's attention to the fact that the hypotheses concerning the general and the specific model do not make any prediction about the reported category, but only about the objective stimulus category (which is illustrated). We state the exact components of each hypothesis and their predictions in methods (note that there is no reported category involved in mathematical notations) and we have further clarified this when we introduce the specific-model hypothesis:

Page 7 top: "the brain state that facilitates recognition of *stimulus from category a* differs from the brain state that facilitates recognition of *stimulus from category b*. This hypothesis predicts that in recognized trials, pre-stimulus activity patterns contain information about the *objective category* of forthcoming stimuli."

Methods:

"Let \mathbb{X} be a set of brain states preceding the appearance of the stimulus from category in set \mathbb{S} and let \mathbb{R} be the set of subjective recognition reports, $\mathbb{X} = \{\mathbf{x}_1, \mathbf{x}_2, \dots, \mathbf{x}_n\}$, $\mathbb{S} = \{\text{face, object, animal, house}\}$, $\mathbb{R} = \{\text{yes, no}\}$ "

"Specific model hypothesis: A subset of brain states \mathbb{X}_a exists ($\mathbb{X}_a \subset \mathbb{X}$) such that the probability of recognizing stimulus $s_a \in \mathbb{S}$ is higher than probability of recognizing stimulus $s_b \in \mathbb{S}$ arriving when the brain is in state $\mathbf{x}_a \in \mathbb{X}_a$ "

We acknowledge that in the first version of the submitted manuscript we were not careful enough to specify "reported" vs "objective" category in all places; we have thoroughly revised the manuscript to ensure correct communication. Figures 2A-C show data testing the two hypotheses, while the following figures/results in the paper provide further insight into how the specific and general models influence various components of perceptual behavior. For example, we ask whether the specific model directly influences the reported category – the answer to this question does not say anything about the model's validity. Similarly, we ask how the two models influence recognition behavior in scrambled-image trials – again the results give more-in-depth insights into the two models' influence on behavior (i.e., specific model influences sensitivity and general model influences criterion). These analyses shed light on the mechanisms and consequences of the discovered ongoing brain processes but are not integral to testing the validity of the general or specific model.

We have added category report to trial type illustration in figure 1D to show the potential influence of prestimulus activity on category reports.

The phrase "multivariate pattern analysis applied to whole-brain activity" in the abstract could more explicitly point to sensor-level topographies, e.g. "multivariate pattern analysis applied across all MEG sensors" or "applied to the MEG sensor-level activity pattern".

Done.